# Avoiding Catastrophe in Online Learning by Asking for Help

## Abstract

Most learning algorithms with formal regret guarantees assume that no mistake is irreparable and essentially rely on trying all possible behaviors. This approach is problematic when some mistakes are *catastrophic*, i.e., irreparable. We propose an online learning problem where the goal is to minimize the chance of catastrophe. Specifically, we assume that the payoff in each round represents the chance of avoiding catastrophe that round and try to maximize the product of payoffs (the overall chance of avoiding catastrophe) while allowing a limited number of queries to a mentor. We first show that in general, any algorithm either constantly queries the mentor or is nearly guaranteed to cause catastrophe. However, in settings where the mentor policy class is learnable in the standard online model, we provide an algorithm whose regret and rate of querying the mentor both approach 0 as the time horizon grows. Conceptually, if a policy class is learnable in the absence of catastrophic risk, it is learnable in the presence of catastrophic risk if the agent can ask for help.

## 1 Introduction

There has been mounting concern over catastrophic risk from AI, including but not limited to autonomous weapon accidents (Abaimov & Martellini, 2020), bioterrorism (Mouton et al., 2024), and cyberattacks on critical infrastructure (Guembe et al., 2022). See Critch & Russell (2023) and Hendrycks et al. (2023) for taxonomies of societal-scale AI risks. In this paper, we use "catastrophe" to refer to any kind of irreparable harm. This definition also covers smaller-scale (yet still unacceptable) incidents such as serious medical errors (Di Nucci, 2019), crashing a robotic vehicle (Kohli & Chadha, 2020), or discriminatory sentencing (Villasenor & Foggo, 2020).

The gravity of these risks contrasts starkly with the dearth of theoretical understanding of how to avoid them. Nearly all of learning theory explicitly or implicitly assumes that no single mistake is too costly. We focus on *online learning*, where an agent repeatedly interacts with an unknown environment and uses its observations to gradually improve its performance. Most online learning algorithms essentially try all possible behaviors and see what works well. We do not want autonomous weapons or surgical robots to try all possible behaviors.

More precisely, trial-and-error-style algorithms only work when catastrophe is assumed to be impossible. This assumption manifests differently in different subtypes of online learning. In the standard online learning model, the agent's actions have no permanent effect on the environment.[1] Online reinforcement learning allows the agent's actions to permanently affect the environment, but typically assumes that either no action has irreversible effects (e.g., Jaksch et al. (2010)) or that the agent is reset at the start of each "episode" (e.g., Azar et al. (2017)). One could train an agent entirely in a controlled lab setting where the above assumptions do hold, but we argue that sufficiently general agents will inevitably encounter novel scenarios when deployed in the real world. Machine learning models often behave unpredictably in unfamiliar environments (see, e.g., Quionero-Candela et al. (2009)), and we do not want AI biologists or robotic vehicles to behave unpredictably.

The goal of this paper is to understand the conditions under which it is possible to formally guarantee avoidance of catastrophe in online learning. Certainly some conditions are necessary, because if the agent can only learn by trying actions directly, the problem is hopeless: any untried action could

---

[1]More precisely, the input can depend on the agent's previous actions, but the agent's performance is always evaluated with respect to the optimal policy on the same sequence of inputs.

lead to paradise or disaster and the agent has no way to predict which. In the real world, however, one needn't learn through pure trial-and-error: one can also ask for help. We think it is critical for high-stakes AI applications to employ a designated supervisor who can be asked for help. Examples include a human doctor supervising AI doctors, a robotic vehicle with a human driver who can take over in emergencies, autonomous weapons with a human operator, and many more. We hope that our work constitutes a small step in the direction of practical safety guarantees for such applications.

## 1.1 OUR MODEL

We propose an online learning model of avoiding catastrophe with mentor help. On each time step, the agent observes an input, selects an action (or queries the mentor), and obtains a payoff. Each payoff represents the probability of avoiding catastrophe on that time step (conditioned on no prior catastrophe). The agent's goal is to maximize the *product* of payoffs, which is equal to the overall probability of avoiding catastrophe by the chain rule of probability.

The (possibly suboptimal) mentor has a fixed policy, and when queried, the mentor illustrates their policy's action for the current input. We desire an agent whose regret – defined as the gap between the mentor's performance and the agent's performance – approaches zero as the time horizon $T$ grows. In other words, with enough time, the agent should avoid catastrophe nearly as well as the mentor. We also expect the agent to become self-sufficient over time: formally, the number of queries to the mentor should be sublinear in $T$, or equivalently, the rate of querying the mentor should go to zero.

## 1.2 OUR ASSUMPTIONS

The agent needs some way to make inferences about unqueried inputs in order to decide when to ask for help. Much past work has used Bayesian inference, which suffers tractability issues in complex environments.[2] We instead assume that the mentor policy satisfies what we call *local generalization*: informally, if the mentor told us that an action was safe for a similar input, then that action is probably also safe for the current input (see Section 3 for a formal definition and further discussion). This captures the intuition that one can transfer knowledge between similar situations. Unlike Bayesian inference, local generalization only requires computing distances and is compatible with any input space which admits a distance metric.

Unlike the standard online learning model, we assume that the agent does not observe payoffs. This is because the payoff in our model represents the chance of avoiding catastrophe on that time step. In the real world, one only observes whether catastrophe occurred, not its probability.[3]

## 1.3 STANDARD ONLINE LEARNING

An overview of standard online learning is in order before discussing our results. In the standard model, the agent observes an input on each time step and must choose an action. An adversary then reveals the correct action, which results in some payoff to the agent. The goal is sublinear regret with respect to the sum of payoffs, or equivalently, the average regret per time step should go to 0 as $T \to \infty$. Figure 1 delineates the precise differences between the standard model and our model.

If the adversary's choices are unconstrained, the problem is hopeless: if the adversary determines the correct action on each time step randomly and independently, the agent can do no better than random guessing. However, sublinear regret becomes possible if (1) the hypothesis class has finite Littlestone dimension (Littlestone, 1988), or (2) the hypothesis class has finite VC dimension (Vapnik & Chervonenkis, 1971) and the input is $\sigma$-*smooth*[4] (Haghtalab et al., 2024).

The goal of sublinear regret in online learning implicitly assumes catastrophe is impossible: the agent can make arbitrarily many (and arbitrarily costly) mistakes as long as the *average* regret per time step goes to 0. In contrast, we demand subconstant regret: the *total* probability of catastrophe should go to 0. Furthermore, standard online learning allows the agent to observe payoffs on every time step, while our agent only receives feedback on time steps with queries. However, access to a mentor (and

---

[2]For the curious reader, Betancourt (2018) provides a thorough treatment. See also Section 2.

[3]One may be able to detect "close calls" in some cases, but observing the precise probability seems unrealistic.

[4]Informally, the adversary chooses a distribution over inputs instead of a precise input. See Section 3.

|  | Objective | Regret goal | Feedback | Mentor | Local gen. |
|---|---|---|---|---|---|
| Standard model | Sum of payoffs | Sublinear | Every time step | No | No |
| Our model | Product of payoffs | Subconstant | Only from queries | Yes | Yes |

Figure 1: Comparison between the standard online learning model and our model.

local generalization) allows our agent to learn without trying actions directly, which is enough to offset all of the above disadvantages.

### 1.4 OUR RESULTS

At a high level, we show that avoiding catastrophe with the help of a mentor and local generalization is no harder than online learning without catastrophic risk.

More precisely, we first show that in general, any algorithm with sublinear queries to the mentor has arbitrarily poor regret in the worst-case (Theorem 4.1). This means that even when the mentor can avoid catastrophe with certainty, any algorithm either needs excessive supervision or is nearly guaranteed to cause catastrophe. Unlike online learning where the general impossibility result is trivial (the agent might as well guess randomly given an unconstrained adversary), local generalization significantly limits the adversary's power and necessitates a careful analysis.

Next, we present a simple algorithm whose total regret and rate of querying the mentor both go to 0 as $T \to \infty$ when either (1) the mentor policy class has finite Littlestone dimension or (2) the mentor policy class has finite VC dimension and the input sequence is $\sigma$-smooth. Our algorithm can handle a multi-dimensional unbounded input space and does not need detailed access to the feature embedding, instead using two simple operations. It does need to know the mentor policy class, as is standard in online learning. We initially prove the theorem for binary actions (Theorem 5.2) and then reduce learning with many actions to the binary action case (Theorem C.1).

Along the way, we prove that the same subconstant bound holds for standard additive regret (Theorem 5.3). Essentially, our techniques are equally effective for maximizing the sum of payoffs and the product of payoffs. We emphasize the multiplicative objective due to our motivation of avoiding catastrophe, but our subconstant additive regret bound may also be of value. In summary, the combination of a mentor and local generalization allows us to reduce the regret by an entire factor of $T$, resulting in subconstant regret (multiplicative or additive) instead of the typical sublinear regret.

The rest of the paper is structured as follows. Section 2 discusses related work. Section 3 formally defines our model. Section 4 presents our negative result for general mentor policies. Section 5 presents our positive result for simple mentor policy classes. Proofs are deferred to the appendix.

## 2 RELATED WORK

**Learning with irreversible costs.** Despite the ubiquity of irreparable/irreversible costs in the real world, theoretical work on this topic remains limited. This may be due to the fundamental modeling question of how the agent should learn about novel inputs or actions without actually trying them.

The most common approach is to allow the agent to ask for help. This alone is insufficient, however: the agent must have some way to decide *when* to ask for help. A popular solution is to perform Bayesian inference on the world model, but this has two tricky requirements: (1) a prior distribution which contains the true world model (or an approximation), and (2) an environment where computing (or approximating) the posterior is tractable. A finite set of possible environments satisfies both conditions, but is unrealistic in many real-world scenarios. In contrast, our algorithm can handle an uncountable policy class and a continuous unbounded input space, which is crucial for many real-world scenarios in which one never sees the exact same input twice.

Bayesian inference combined with asking for help is studied by Cohen et al. (2021); Cohen & Hutter (2020); Kosoy (2019); Mindermann et al. (2018). We also mention Hadfield-Menell et al. (2017); Moldovan & Abbeel (2012); Turchetta et al. (2016), which utilize Bayesian inference in the context of safe (online) reinforcement learning without asking for help (and without regret bounds).

We are only aware of two papers which theoretically address irreversibility without Bayesian inference: Grinsztajn et al. (2021) and Maillard et al. (2019). The former proposes to sample trajectories and learn reversibility based on temporal consistency between states: intuitively, if $s_1$ always precedes $s_2$, we can infer that $s_1$ is unreachable from $s_2$. Although the paper theoretically grounds this intuition, there is no formal regret guarantee. The latter presents an algorithm which asks for help in the form of rollouts from the current state. However, the regret bound and number of rollouts are both linear in the worst case, due to the dependence on the $\gamma^*$ parameter which roughly captures how bad an irreversible action can be. In contrast, our algorithm achieves good regret even when actions are maximally bad.

To our knowledge, we are the first to provide an algorithm which formally guarantees avoidance of catastrophe (with high probability) without Bayesian inference. We are also not aware of prior results comparable to our negative result, including in the Bayesian regime.

**Constrained Markov Decision Processes (CMDPs).** CMDPs (Altman, 2021; Puterman, 2014) require the agent to maximize reward while also satisfying safety constraints. The two most relevant papers are Liu et al. (2021) and Stradi et al. (2024), both of which provide algorithms guaranteed to satisfy initially unknown safety constraints with high probability on every time step. However, both papers assume that the agent knows a fully safe policy upfront. In contrast, the agent in our setting has no prior knowledge. In this sense, our work complements theirs: our goal is essentially to learn the baseline safe policy that their algorithms require. One can also view our problem as the "pessimistic" model and their problem as the "optimistic" model, with some applications better captured by our model while other applications are better captured by theirs.

**Online learning.** See Cesa-Bianchi & Lugosi (2006) and Chapter 21 of Shalev-Shwartz & Ben-David (2014) for introductions to online learning. A classical result states that sublinear regret is possible iff the hypothesis class has finite Littlestone dimension (Littlestone, 1988). However, even some simple hypothesis classes have infinite Littlestone dimension, such as the class of thresholds on $[0, 1]$ (Example 21.4 in Shalev-Shwartz & Ben-David (2014)). Recently, Haghtalab et al. (2024) showed that if the adversary only chooses a distribution over inputs rather than the precise input, only the weaker assumption of finite VC dimension (Vapnik & Chervonenkis, 1971) is needed for sublinear regret. Specifically, they assume that each input is sampled from a distribution whose concentration is upper bounded by $\frac{1}{\sigma}$ times the uniform distribution. This framework is known as *smoothed analysis*, originally proposed by Spielman & Teng (2004).

**Multiplicative objectives.** Although online learning traditionally studies the sum of payoffs, there is some work on maximizing the product of payoffs, or equivalently the sum of logarithms (Chapter 9 of Cesa-Bianchi et al. (2017)). However, these regret bounds are still sublinear in $T$, in comparison to our subconstant regret bounds. Also, that work still assumes that payoffs are observed on every time step, while our agent only receives feedback in response to queries (Figure 1).

Barman et al. (2023) recently provided regret bounds for a multiplicative objective in a multi-armed bandit problem, but their objective is the geometric mean of payoffs instead of the product. Interpreted in our context, their regret bounds imply that the *average* chance of catastrophe goes to zero, while we guarantee that the *total* chance of catastrophe goes to zero. This distinction is closed related to the difference between subconstant and sublinear regret discussed in Section 1.3.

**Active learning and imitation learning.** Our assumption that the agent only receives feedback in response to queries falls under the umbrella of active learning (Hanneke, 2014). This contrasts with passive learning, where the agent receives feedback automatically. Although ideas from active learning could be useful in our domain, we are not aware of any results from that literature which account for irreversible costs. The process of the agent learning from a mentor is also reminiscent of imitation learning (Osa et al., 2018), but we are not aware of any relevant technical implications.

## 3   MODEL

**Inputs.** Let $\mathbb{N}$ refer to the set of strictly positive integers and let $T \in \mathbb{N}$ be the time horizon. Let $\mathcal{X} \subseteq \mathbb{R}^n$ be the input space[5] and let $\mathbf{x} = (x_1, x_2, \ldots x_T) \in \mathcal{X}^T$ be the sequence of inputs. In the adversarial setting, each $x_t$ can have arbitrary dependence on the events of prior time steps. In the smoothed setting, the adversary only chooses the distribution $x_t$ from which $x_t$ is sampled. Formally,

---

[5]One could also allow a generic metric space; our assumption of $\mathcal{X} \subseteq \mathbb{R}^n$ is only for convenience.

a distribution $\mathcal{D}$ over $\mathcal{X}$ is $\sigma$-smooth if for any $X \subseteq \mathcal{X}$, $\mathcal{D}(X) \leq \frac{1}{\sigma}U(X)$. (In the smoothed setting, we assume that $\mathcal{X}$ supports a uniform distribution $U$.[6]) If each $x_t$ is sampled from a $\sigma$-smooth $\mathcal{D}_t$, we say that $\mathbf{x}$ is $\sigma$-smooth. The sequence $\boldsymbol{\mathcal{D}} = \mathcal{D}_1, \ldots, \mathcal{D}_T$ can still be adaptive, i.e., the choice of $\mathcal{D}_t$ can depend on the events of prior time steps.

**Actions.** Let $\mathcal{Y}$ be a finite set of actions. There also exists a special action $\tilde{y}$ which corresponds to querying the mentor. For $k \in \mathbb{N}$, let $[k] = \{1, 2, \ldots, k\}$. On each time step $t \in [T]$, the agent must select an action $y_t \in \mathcal{Y} \cup \{\tilde{y}\}$, which generates payoff $\mu(x_t, y_t) \in [0, 1]$. Unless otherwise noted, all expectations are over the agent's randomization (if any) and the randomization in $\mathbf{x}$ (if any).

**Asking for help.** The mentor is endowed with a (not necessarily optimal) policy $\pi^m : \mathcal{X} \to \mathcal{Y}$. When action $\tilde{y}$ is chosen, the mentor informs the agent of the action $\pi^m(x_t)$ and the agent obtains payoff $\mu(x_t, \pi^m(x_t))$. For brevity, let $\mu^m(x) = \mu(x, \pi^m(x))$. The agent never observes payoffs: the only way to learn about $\mu$ is by querying the mentor.

We would like an algorithm which becomes "self-sufficient" over time: the rate of querying the mentor should go to 0 as $T \to \infty$, or equivalently, the cumulative number of queries should be sublinear in $T$. Formally, let $Q_T(\mu, \pi^m) = \{t \in [T] : y_t = \tilde{y}\}$ be the random variable denoting the set of time steps with queries. Then we say that the (expected) number of queries is sublinear in $T$ if $\sup_{\mu, \pi^m} \mathbb{E}[|Q_T(\mu, \pi^m)|] \in o(T)$. In other words, there must exist $g : \mathbb{N} \to \mathbb{N}$ such that $g(T) \in o(T)$ and $\sup_{\mu, \pi^m} \mathbb{E}[|Q_T(\mu, \pi^m)|] \leq g(T)$.[7] For brevity, we will usually write $Q_T = Q_T(\mu, \pi^m)$.

**Local generalization.** We assume that the mentor policy permits "local generalization". Informally, if the agent is given an input $x$, taking the mentor action for a similar input $x'$ is almost as good as taking the mentor action for $x$. Formally, we assume there exists $L > 0$ such that for all $x, x' \in \mathcal{X}$, $|\mu^m(x) - \mu(x, \pi^m(x'))| \leq L\|x - x'\|$, where $\|\cdot\|$ denotes Euclidean distance. This represents the ability to transfer knowledge between similar inputs:

$$| \underbrace{\mu(x, \pi^m(x))}_{\text{Taking the "right" action}} - \underbrace{\mu(x, \pi^m(x'))}_{\text{Using what you learned in } x'} | \leq \underbrace{L\|x - x'\|}_{\text{Similarity between } x \text{ and } x'}$$

Borrowing knowledge from similar experiences seems fundamental to learning and is well-understood in the psychology literature (Esser et al., 2023) and education literature (Hajian, 2019).

Crucially, our input space can be any feature embedding of the agent's situation, not just its physical positioning. Our algorithms will not require knowledge of the feature embedding and do not need to know $L$, so it suffices that there exists *some* feature embedding which satisfies local generalization. The agent does not even need to know which embedding it is. Finally, local generalization implies the more familiar Lipschitz continuity for an optimal mentor (Proposition E.1).

**Multiplicative objective and regret.** If $\mu(x_t, y_t) \in [0, 1]$ is the chance of avoiding catastrophe on time step $t$ (conditioned on no prior catastrophe), then $\prod_{t=1}^{T} \mu(x_t, y_t)$ is the agent's overall chance of avoiding catastrophe.[8] For a fixed $\mathbf{x}$ and agent actions $\mathbf{y} = (y_1, \ldots, y_T)$, the agent's *regret* is

$$R_T(\mathbf{x}, \mathbf{y}, \mu, \pi^m) = \prod_{t=1}^{T} \mu^m(x_t) - \prod_{t=1}^{T} \mu(x_t, y_t)$$

We will usually write $R_T = R_T(\mathbf{x}, \mathbf{y}, \mu, \pi^m)$ for brevity. We will study the expected regret over any randomness in $\mathbf{x}$ and/or $\mathbf{y}$. We desire subconstant worst-case regret: the total (not average) expected regret should go to 0 for any $\mu$ and $\pi^m$. Formally, we want $\lim_{T \to \infty} \sup_{\mu, \pi^m} \mathbb{E}[R_T] = 0$.

The value of a bound on $\mathbb{E}[R_T]$ depends on the quality of the mentor. In particular, subconstant regret becomes trivial if $\lim_{T \to \infty} \mathbb{E}\left[\prod_{t=1}^{T} \mu^m(x_t)\right] = 0$. However, we think that high-stakes AI applications should ensure the presence of a mentor who is almost always safe, i.e., $\mathbb{E}\left[\prod_{t=1}^{T} \mu^m(x_t)\right] \approx 1$.

---

[6]For example, $\mathcal{X}$ having finite Lebesgue measure is sufficient. Note that this does not imply boundedness. Alternatively, $\sigma$-smoothness can be defined with respect to a different distribution, as long as the Radon-Nikodym derivative is uniformly bounded; see Definition 1 of Block et al. (2022).

[7]One could instead consider the worst-case number of queries, but this distinction does not affect whether subconstant regret is achievable (Proposition E.2).

[8]Conditioning on no prior catastrophe means we do not need to assume that these probabilities are independent (and if catastrophe has already occurred, this time step does not matter). This is due to the chain rule of probability.

If no such mentor exists for some application, perhaps it is better to avoid the application altogether. Also, our regret bounds include rates of convergence, so even if the mentor policy is guaranteed to eventually cause catastrophe, we can still bound how quickly the agent becomes unsafe.

**VC and Littlestone dimensions.** VC dimension (Vapnik & Chervonenkis, 1971) and Littlestone dimension (Littlestone, 1988) are standard measures of learning difficulty which capture the ability of a hypothesis class (in our case, a policy class) to realize arbitrary combinations of labels (in our case, actions). We omit the precise dimensions since we only utilize these concepts via existing results. See Shalev-Shwartz & Ben-David (2014) for a comprehensive overview.

**Misc.** The diameter of a set $X \subseteq \mathcal{X}$ is defined by $\text{diam}(X) = \max_{x,x' \in X} ||x - x'||$. All logarithms and exponents are base $e$ unless otherwise noted.

## 4 AVOIDING CATASTROPHE IS IMPOSSIBLE IN GENERAL

We begin by showing that in general, any algorithm with sublinear mentor queries has arbitrary poor regret in the worst-case, even when inputs are i.i.d. on $[0, 1]$. The result also holds even if the algorithm knows $L$ and $\mathbf{x}$ ahead of time.

**Theorem 4.1.** *The worst-case expected regret of any algorithm with sublinear queries goes to 1 as $T$ goes to infinity. Formally, $\lim_{T \to \infty} \sup_{\mu, \pi^m} \mathbb{E}[R_T] = 1$.*

### 4.1 INTUITION

We partition $\mathcal{X}$ into equally-sized sections that are "independent" in the sense that querying an input in section $i$ gives you no information about section $j$. The number of sections is determined by a function $f : \mathbb{N} \to \mathbb{N}$ that we will choose. If $|Q_T| \in o(f(T))$, most of these sections will never contain a query. When the agent sees an input in a section not containing a query, it essentially has to guess, meaning it will be wrong a constant fraction of the time. Figure 2 fleshes out this idea.

**Picking $f(T)$.** A natural idea is to try $f(T) = T$, but this doesn't quite work: even if the agent chooses wrong on every time step, the minimum payoff is still at least $1 - \frac{L}{2T}$, and $\lim_{T \to \infty} \prod_{t=1}^{T} \left(1 - \frac{L}{2T}\right) = \lim_{T \to \infty} \left(1 - \frac{L}{2T}\right)^T = e^{-L/2}$. In order for the regret to approach 1, we need $f(T)$ to be asymptotically between $|Q_T|$ and $T$ (such $f$ must exist since $|Q_T| \leq g(T) \in o(T)$). This leads to the following bound: $\prod_{t=1}^{T} \mu(x_t, y_t) \leq \left(1 - \frac{L}{\Theta(f(T))}\right)^{\Theta(T)}$. When $f(T) \in o(T)$, the right hand side converges to 0, while $\prod_{t=1}^{T} \mu^m(x_t) = 1$. In words, the agent is nearly guaranteed to cause catastrophe, despite the existence of a policy which is guaranteed to avoid catastrophe.

**VC dimension.** The class of mentor policies induced by our construction has VC dimension $f(T)$; considered over all possible values of $T$, this implies infinite VC (and Littlestone) dimension. This is necessary given our positive results in Section 5.

### 4.2 FORMAL DEFINITION OF CONSTRUCTION

Let $\mathcal{X} = [0, 1]$ and $\mathcal{D}_t = U$ for each $t \in [T]$. Assume that $L \leq 1$; this will simplify the math and only makes the problem easier for the agent. We define a family of payoff functions parameterized by a function $f : \mathbb{N} \to \mathbb{N}$ and a bit string $\mathbf{a} = (a_1, a_2, \ldots, a_{f(T)}) \in \{0, 1\}^{f(T)}$. The bit $a_j$ will denote the optimal action in section $j$. Note that $f(T) \geq 1$ and since we defined $\mathbb{N}$ to exclude 0.

For each $j \in [f(T)]$, we refer to $X_j = \left[\frac{j-1}{f(T)}, \frac{j}{f(T)}\right]$ as the $j$th section. Let $m_j = \frac{j - 0.5}{f(T)}$ be the midpoint of $X_j$. Assume that each $x_t$ belongs to exactly one $X_j$ (this happens with probability 1, so this assumption does affect the expected regret). Let $j(x)$ denote the index of the section containing input $x$. Then $\mu_{f,\mathbf{a}}$ is defined by

$$\mu_{f,\mathbf{a}}(x, y) = \begin{cases} 1 & \text{if } y = a_{j(x)} \\ 1 - L\left(\frac{1}{2f(T)} - |m_{j(x)} - x|\right) & \text{if } y \neq a_{j(x)} \end{cases}$$

Let $\pi^m$ be any policy which is optimal for $\mu_{f,\mathbf{a}}$. Note that there is a unique optimal action for each $x_t$, since each $x_t$ belongs to exactly one $X_j$; formally, $\pi^m(x_t) = a_{j(x_t)}$.

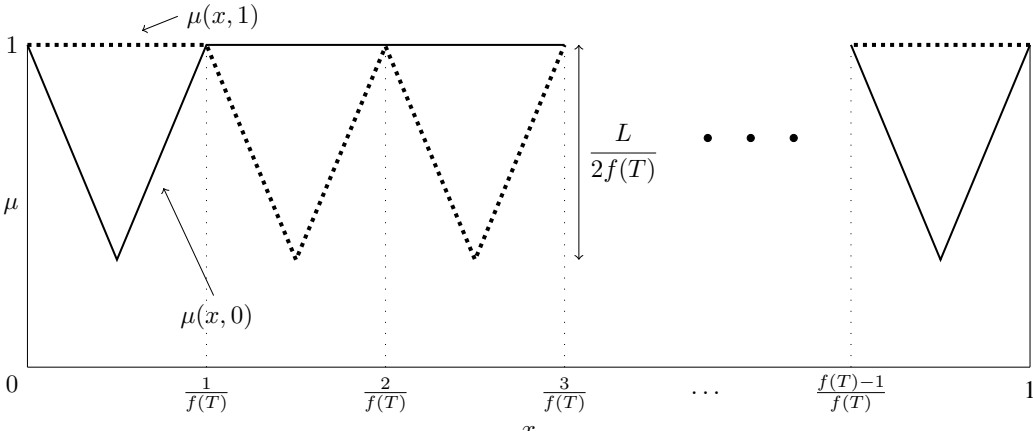

Figure 2: An illustration of the construction we use to prove Theorem 4.1 (not to scale). The horizontal axis indicates the input $x \in [0, 1]$ and the vertical axis indicates the payoff $\mu(x, y) \in [0, 1]$. The solid line represents $\mu(x, 0)$ and the dotted line represents $\mu(x, 1)$. In each section, one of the actions has the optimal payoff of 1, and the other action has the worst possible payoff allowed by $L$, reaching a minimum of $1 - \frac{L}{2f(T)}$. Crucially, both actions result in a payoff of 1 at the boundaries between sections: this allows us to "reset" for the next section. As a result, we can freely toggle the optimal action for each section independently.

For any $\mathbf{a} \in \{0, 1\}^{f(T)}$, $\mu_{f,\mathbf{a}}$ is piecewise linear (trivially) and continuous (because both actions have payoff 1 on the boundary between sections). Since the slope of each piece is in $\{-L, 0, L\}$, $\mu_{f,\mathbf{a}}$ is Lipschitz continuous. Thus by Proposition E.1, $\pi^m$ satisfies local generalization.

# 5 AVOIDING CATASTROPHE ASSUMING FINITE VC OR LITTLESTONE DIMENSION

Theorem 4.1 shows that avoiding catastrophe is impossible in general, which is also true in online learning. What if we restrict ourselves to settings where standard online learning is possible? Specifically, we assume that $\pi^m$ belongs to a policy class $\Pi$ where either (1) $\Pi$ has finite VC dimension $d$ and $\mathbf{x}$ is $\sigma$-smooth or (2) $\Pi$ has finite Littlestone dimension $d$.[9]

This section presents a simple algorithm which guarantees subconstant regret and sublinear queries under either of those assumptions. Our algorithm needs to know $\Pi$, as is standard in online learning. The algorithm does not need to know $\sigma$ (in the smooth case) or $L$, and can handle an unbounded input space (the number of queries simply scales with the maximum distance between observed inputs).

For simplicity, we initially prove our result for $\mathcal{Y} = \{0, 1\}$. Appendix C extends our result to many actions using the standard "one versus rest" reduction.[10]

## 5.1 INTUITION BEHIND THE ALGORITHM

Algorithm 1 has two simple components: (1) run a modified version of the Hedge algorithm for online learning, but (2) ask for help for unfamiliar inputs (specifically, when the current input is very different from any queried input with the same action under the proposed policy). Hedge ensures that the number of time steps where the agent's action doesn't match the mentor's is small, and asking for help for unfamiliar inputs ensures that when we do make a mistake, the cost isn't too high. This algorithmic structure seems quite natural: mostly follow a baseline strategy, but ask for help when out-of-distribution.

**Simple operations.** The algorithm does not require detailed access to the input embedding, instead relying on two simple operations: evaluating a policy on a particular input, and computing a nearest

---

[9]Recall from Section 1.3 that standard online learning becomes tractable under either of these assumptions.

[10]For each action $y$, we learn a binary classifier which predicts whether $\pi^m(x) = y$. If every binary classifier is correct, we can correctly determine $\pi^m(x)$. See, e.g., Chapter 29 of Shalev-Shwartz & Ben-David (2014).

**Algorithm 1** successfully avoids catastrophe assuming finite VC or Littlestone dimension.

1: **function** AVOIDCATASTROPHE($T \in \mathbb{N}$, $\varepsilon \in \mathbb{R}_{>0}$, $d \in \mathbb{N}$, policy class $\Pi$)
2:     **if** $\Pi$ has VC dimension $d$ **then**
3:         $\tilde{\Pi} \leftarrow$ any smooth $\varepsilon$-cover of $\Pi$ of size at most $(41/\varepsilon)^d$          ▷ See Definition 5.4
4:     **else**
5:         $\tilde{\Pi} \leftarrow$ any adversarial cover of size at most $(eT/d)^d$          ▷ See Definition 5.5
6:     $X \leftarrow \emptyset$
7:     $w(\pi) \leftarrow 1$ for all $\pi \in \tilde{\Pi}$
8:     $p \leftarrow 1/\sqrt{\varepsilon T}$
9:     $\eta \leftarrow \max\left(\sqrt{\frac{p \log |\tilde{\Pi}|}{2T}}, \frac{p^2}{\sqrt{2}}\right)$
10:     **for** $t$ **from** 1 **to** $T$ **do**          ▷ Run one step of Hedge, which selects policy $\pi_t$
11:         hedgeQuery $\leftarrow$ true with probability $p$ else false
12:         **if** hedgeQuery **then**
13:             Query mentor and observe $\pi^m(x_t)$
14:             $\ell(t, \pi) \leftarrow \mathbf{1}(\pi(x_t) \neq \pi^m(x_t))$ for all $\pi \in \tilde{\Pi}$
15:             $\ell^* \leftarrow \min_{\pi \in \tilde{\Pi}} \ell(t, \pi)$
16:             $w(\pi) \leftarrow w(\pi) \cdot \exp(-\eta(\ell(t, \pi) - \ell^*))$ for all $\pi \in \tilde{\Pi}$
17:             $\pi_t \leftarrow \arg\min_{\pi \in \tilde{\Pi}} \ell(t, \pi)$
18:         **else**
19:             $P(\pi) \leftarrow w(\pi)/\sum_{\pi' \in \tilde{\Pi}} w(\pi')$ for all $\pi \in \tilde{\Pi}$
20:             Sample $\pi_t \sim P$
21:         **if** $\min_{(x,y) \in X : y = \pi_t(x_t)} ||x_t - x|| > \varepsilon^{1/n}$ **then**     ▷ Ask for help if out-of-distribution
22:             Query mentor and observe $\pi^m(x_t)$
23:             $X \leftarrow X \cup \{(x_t, \pi^m(x_t))\}$
24:         **else**          ▷ Otherwise, follow Hedge's chosen policy
25:             Take action $\pi_t(x_t)$

neighbor distance. The former seems necessary for any algorithm. The latter could be modeled as an out-of-distribution detector score, for which many methods are available (see e.g., Yang et al. (2024)).

**Hedge.** Hedge (Freund & Schapire, 1997) is a standard online learning algorithm which ensures sublinear regret when the number of hypotheses (in our case, the number of policies in $\Pi$) is finite.[11] We would prefer not to assume that $\Pi$ is finite. Luckily, any policy $\Pi$ can be approximated within $\varepsilon$ when either (1) $\Pi$ has finite VC dimension and $\mathbf{x}$ $\sigma$-smooth or (2) $\Pi$ has finite Littlestone dimension. Thus we can run Hedge on this approximative policy class instead.

One other modification is necessary. In standard online learning, losses are observed on every time step, but our agent only receives feedback in response to queries. To handle this, we modify Hedge to only perform updates on time steps with queries and to issue a query with probability $p$ on each time step. Continuing our lucky streak, Russo et al. (2024) analyzes exactly this modification of Hedge.

We prove the following theorem parametrized by $\varepsilon$:

**Theorem 5.1.** *Let $\mathcal{Y} = \{0, 1\}$. Assume $\pi^m \in \Pi$ where either (1) $\Pi$ has finite VC dimension $d$, $\mathbf{x}$ is $\sigma$-smooth, and $\varepsilon T \log T > 12\sigma d \log(4e^2/\varepsilon)$ or (2) $\Pi$ has finite Littlestone dimension $d$. Then for any $T \in \mathbb{N}$ and $\varepsilon > 0$, Algorithm 1 satisfies*

$$\mathbb{E}[R_T] \in O\left(\frac{dL}{\sigma} T \varepsilon^{1+1/n} \log(1/\varepsilon) \log T\right)$$

$$\mathbb{E}[|Q_T|] \in O\left(\sqrt{\frac{T}{\varepsilon}} + \frac{d}{\sigma} T \varepsilon \log(1/\varepsilon) \log T + \frac{\text{diam}(\mathbf{x})^n}{\varepsilon}\right)$$

In Case 1, the expectation is over the randomness of both $\mathbf{x}$ and the algorithm, while in Case 2, the expectation is over only the randomness of the algorithm. Also, $R_T$ and $Q_T$ clearly have no dependence on $\sigma$ in Case 2, but we include $\sigma$ anyway to avoid writing two separate bounds.

---

[11]See Chapter 5 of Slivkins et al. (2019) and Chapter 21 of Shalev-Shwartz & Ben-David (2014) for modern introductions to Hedge.

To obtain subconstant regret and sublinear queries, we can choose $\varepsilon = T^{\frac{-2n}{2n+1}}$. This also satisfies the requirement of $\varepsilon T \log T > 12\sigma d \log(4e^2/\varepsilon)$ for large enough $T$.

**Theorem 5.2.** *Let* $\mathcal{Y} = \{0, 1\}$. *Assume* $\pi^m \in \Pi$ *where either (1)* $\Pi$ *has finite VC dimension* $d$ *and* $\mathbf{x}$ *is* $\sigma$-*smooth or (2)* $\Pi$ *has finite Littlestone dimension* $d$. *Then for any* $T \in \mathbb{N}$, *Algorithm 1 with* $\varepsilon = T^{\frac{-2n}{2n+1}}$ *satisfies*

$$\mathbb{E}\left[R_T\right] \in O\left(\frac{dL}{\sigma} T^{\frac{-1}{2n+1}} \log T\right)$$

$$\mathbb{E}[|Q_T|] \in O\left(T^{\frac{4n+1}{4n+2}} \left(\frac{d}{\sigma} \log T + \mathrm{diam}(\mathbf{x})^n\right)\right)$$

Although our focus is the product of payoffs, Algorithm 1 also guarantees subconstant additive regret:

**Theorem 5.3.** *Let* $\mathcal{Y} = \{0, 1\}$. *Assume* $\pi^m \in \Pi$ *where either (1)* $\Pi$ *has finite VC dimension* $d$ *and* $\mathbf{x}$ *is* $\sigma$-*smooth or (2)* $\Pi$ *has finite Littlestone dimension* $d$. *Then for any* $T \in \mathbb{N}$, *Algorithm 1 with* $\varepsilon = T^{\frac{-2n}{2n+1}}$ *satisfies*

$$\mathbb{E}\left[\sum_{t=1}^{T} \mu^m(x_t) - \sum_{t=1}^{T} \mu(x_t, y_t)\right] \in O\left(\frac{dL}{\sigma} T^{\frac{-1}{2n+1}} \log T\right)$$

## 5.2 PROOF SKETCH

The formal proof of Theorem 5.1 can be found in Appendix B, but we outline the key elements here. The regret analysis consists of two ingredients: analyzing the Hedge component, and analyzing the "ask for help when out-of-distrubtion" component. The former will bound the number of mistakes made by the algorithm (i.e., the number of time steps where the agent's action doesn't match the mentor's), and the latter will bound the cost of any single mistake. We must also carefully show that the latter does not result in excessively many queries, which we do via a novel packing argument.

We begin by formalizing two notion of approximating a policy class:

**Definition 5.4.** Let $U$ be the uniform distribution over $\mathcal{X}$. For $\varepsilon > 0$, a policy class $\tilde{\Pi}$ is a *smooth* $\varepsilon$-*cover* of a policy class $\Pi$ is for every $\pi \in \Pi$, there exists $\tilde{\pi} \in \tilde{\Pi}$ such that $\Pr_{x \sim U}[\pi(x) \neq \tilde{\pi}(x)] \leq \varepsilon$.

**Definition 5.5.** A policy class $\tilde{\Pi}$ is an *adversarial cover* of a policy class $\Pi$ is for every $\mathbf{x} \in \mathcal{X}^T$ and $\pi \in \Pi$, there exists $\tilde{\pi} \in \tilde{\Pi}$ such that $\pi(x_t) = \tilde{\pi}(x_t)$ for all $t \in [T]$.

The existence of small covers is crucial:

**Lemma 5.1** (Lemma 7.3.2 in Haghtalab (2018)[12]). *For all* $\varepsilon > 0$, *any policy class of VC dimension* $d$ *admits a smooth* $\varepsilon$-*cover of size at most* $(41/\varepsilon)^d$.

**Lemma 5.2** (Lemmas 21.13 and A.5 in Shalev-Shwartz & Ben-David (2014)). *Any policy class of Littlestone dimension* $d$ *admits an adversarial cover of size at most* $(eT/d)^d$.

An adversarial cover is a perfect cover by definition. The following lemma establishes that a smooth $\varepsilon$-cover is a good approximation for any sequence of $\sigma$-smooth distributions.

**Lemma 5.3** (Equation 2 and Lemma 3.3 in Haghtalab et al. (2024)). *Let* $\tilde{\Pi}$ *be a finite smooth* $\varepsilon$-*cover of* $\Pi$ *and let* $\mathcal{D} = \mathcal{D}_1, \ldots, \mathcal{D}_T$ *be a sequence of* $\sigma$-*smooth distributions. If* $\varepsilon T \log T > 12\sigma d \log(4e^2/\varepsilon)$, *then* $\mathbb{E}_{\mathbf{x} \sim \mathcal{D}}\left[\sup_{\pi \in \Pi} \min_{\tilde{\pi} \in \tilde{\Pi}} \sum_{t=1}^{T} \mathbf{1}(\pi(x_t) \neq \tilde{\pi}(x_t))\right] \in O\left(\frac{1}{\sigma} T \varepsilon \log T \sqrt{d \log(1/\varepsilon)}\right)$.

We will run a variant of Hedge on $\tilde{\Pi}$. The vanilla Hedge algorithm operates in the standard online learning model where on each time step, the agent selects a policy (or more generally, a hypothesis), and observes the *loss* of every policy. In general the loss function can depend arbitrarily on the time step, the policy, and prior events, but we will only use the indicator loss function $\ell(t, \pi) = \mathbf{1}(\pi(x_t) \neq \pi^m(x_t))$. Crucially, whenever we query and learn $\pi^m(x_t)$, we can compute $\ell(t, \pi)$ for every $\pi \in \tilde{\Pi}$.

---

[12]See also Haussler & Long (1995) or Lemma 13.6 in Boucheron et al. (2013) for variants which are less convenient for our purposes.

We cannot afford to query on every time step, however. Recently, Russo et al. (2024) analyzed a variant of Hedge where losses are observed only in response to queries, which they call "label-efficient feedback". They proved a regret bound when a query is issued on each time step with fixed probability $p$. Lemma 5.4 restates their result in a form that is more convenient for us (see Appendix B for details). Although their result is stated for non-adaptive adversaries, we explain in Appendix B.3 how their argument easily generalizes to adaptive adversaries. Full pseudocode for HEDGEWITHQUERIES can also be found in the appendix (Algorithm 2).

**Lemma 5.4** (Lemma 3.5 in Russo et al. (2024)). *Assume* $\tilde{\Pi}$ *is finite. Then for any loss function* $\ell : [T] \times \tilde{\Pi} \to [0, 1]$ *and query probability* $p$, HEDGEWITHQUERIES *enjoys the regret bound*

$$\sum_{t=1}^{T} \mathbb{E}[\ell(t, \pi_t)] - \min_{\tilde{\pi} \in \tilde{\Pi}} \sum_{t=1}^{T} \ell(t, \pi) \leq \frac{2 \log |\tilde{\Pi}|}{p^2}$$

*where* $\pi_t$ *is the policy chosen at time* $t$ *and the expectation is over the randomness of the algorithm.*

We apply Lemma 5.4 with $\ell(t, \pi) = \mathbf{1}(\pi(x_t) \neq \pi^m(x_t))$ and combine this with Lemmas 5.1 and 5.3 (in the $\sigma$-smooth case) and with Lemma 5.2 (in the adversarial case). This yields a $O\left(\frac{d}{\sigma} T \varepsilon \log(1/\varepsilon) \log T\right)$ bound on the number of mistakes made by Algorithm 1 (Lemma B.1).

The other key ingredient of the proof is analyzing the "ask for help when out-of-distribution" component. Combined with the local generalization assumption, this allows us to fairly easily bound the cost of a single mistake (Lemma B.2). The trickier part is bounding the number of resulting queries. It is tempting to claim that the inputs queried in the out-of-distribution case must all be separated by at least $\varepsilon^{1/n}$ and thus form an $\varepsilon^{1/n}$-packing, but this is actually not true. Instead, we provide a novel method for bounding the number of data points (i.e., queries) needed to cover a set *with respect to the realized actions of the algorithm* (Lemma B.7). This is in contrast to vanilla packing arguments which consider all data points in aggregate. Our method may be useful in other contexts where a more refined packing argument is needed.

## 6 CONCLUSION AND FUTURE WORK

In this paper, we proposed a model of avoiding catastrophe in online learning. We showed that achieving subconstant regret in our problem (with the help of a mentor and local generalization) is no harder than achieving sublinear regret in standard online learning.

There remain some technical questions within this paper's model. One question is whether the time complexity of Algorithm 1 be improved, which currently stands at $\Omega(|\tilde{\Pi}| \cdot T)$ plus the time to compute the $\varepsilon$-cover. Also, we have not resolved whether our problem is tractable for finite VC dimension and fully adversarial inputs (although Appendix D shows that the problem is tractable for at least some classes with finite VC but infinite Littlestone dimension).

We are also interested in alternatives to the local generalization assumption. We should expect some assumption to be necessary: if not, the payoff function $\mu(x, y) = \mathbf{1}(\pi^m(x) = y)$ means the agent essentially has to make zero mistakes, which turns out to be impossible even for $\sigma$-smooth $\mathbf{x}$ and finite VC dimension (Theorem E.3). One possible alternative is Bayesian inference. We intentionally avoided Bayesian approaches in this paper due to tractability concerns, but it seems premature to abandon those ideas entirely.

Finally, we are excited to apply the ideas in this paper to Markov Decision Processes (MDPs): specifically, MDPs where some actions are irreversible ("non-communicating") and the agent only gets one attempt ("single-episode"). In such MDPs, the agent must not only avoid catastrophe but also obtain high reward. As discussed in Section 2, very little theory exists for RL in non-communicating single-episode MDPs. Can an agent learn near-optimal behavior in high-stakes environments while becoming self-sufficient over time? Formally, we pose the following open problem:

*Is there an algorithm for non-communicating single-episode undiscounted MDPs which ensures that both the regret and the number of mentor queries are sublinear in* $T$?

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

## A PROOF OF THEOREM 4.1

### A.1 PROOF ROADMAP

Throughout the proof, let $V_j$ be the set of time steps $t \leq T$ where $|m_j - x_t| \leq \frac{1}{4f(T)}$. In words, $x_t$ is relatively close to the midpoint of $X_j$. This will imply that the suboptimal action is in fact quite suboptimal. This also implies that $x_t$ is in $X_j$, since each $X_j$ has length $1/f(T)$.

The proof proceeds via the following steps:

1. Prove that $f(T) = \sqrt{(|Q_T| + 1)T}$ is asymptotically between $|Q_T|$ and $T$ (Lemma A.1).

2. Provide a simple variant of the Chernoff bound which we will apply multiple times (Lemma A.2).

3. Show that with high probability, $\sum_{j \in A} |V_j|$ is adequately large (Lemma A.3).

4. The key lemma is Lemma A.4, which shows that a randomly sampled **a** produces poor agent performance with high probability. The central idea is that at least $f(T) - |Q_T|$ sections are never queried (which is large, by Lemma A.1), so the agent has no way of knowing the optimal action in those sections. As a result, the agent picks the wrong answer at least half the time on average (and at least a quarter of the time with high probability). Lemma A.3 implies that a constant fraction of those time steps will have quite suboptimal payoffs, again with high probability.

5. Finally, $\sup_{\mu} \mathbb{E}_{\mathbf{x} \sim U^T, \mathbf{y}} R_T(\mathbf{x}, \mathbf{y}, \mu, \pi^m) \geq \mathbb{E}_{\mathbf{a} \sim U(\{0,1\}^{f(T)})} \mathbb{E}_{\mathbf{x} \sim U^T, \mathbf{y}} R_T(\mathbf{x}, \mathbf{y}, \mu_{f,\mathbf{a}}, \pi^m)$, where $U(\{0,1\}^{f(T)})$ is the uniform distribution over bit strings of length $f(T)$. This is essentially an application of the probabilistic method: if a randomly chosen $\mu_{f,\mathbf{a}}$ has high expected regret, then the worst case $\mu$ also has high expected regret.

Note that $\mathbf{x}, \mathbf{y}$, and $\mathbf{a}$ are random variables, so all variables defined on top of them (xuch as $V_j$) are also random variables. In contrast, the partition $\mathcal{X} = \{X_1, \ldots, X_{f(T)}\}$ and properties thereof (like the midpoints $m_j$) are not random variables.

Lastly, while the intuition provided in Section 4.1 is accurate, the analysis will mostly occur in log space, so the bounds will look different. However, bounds of the form discussed in Section 4.1 can still be found as an intermediate step in Part 4 of the proof of Lemma A.4.

### A.2 PROOF

**Lemma A.1.** *Let $a, b : \mathbb{N} \to \mathbb{N}$ be functions such that $a(x) \in o(b(x))$. Then $c(x) = \sqrt{a(x)b(x)}$ satisfies $a(x) \in o(c(x))$ and $c(x) \in o(b(x))$.*

*Proof.* Since $a$ and $b$ are strictly positive (and thus $c$ is as well), we have

$$\frac{a(x)}{c(x)} = \frac{a(x)}{\sqrt{a(x)b(x)}} = \sqrt{\frac{a(x)}{b(x)}} = \frac{\sqrt{a(x)b(x)}}{b(x)} = \frac{c(x)}{b(x)}$$

Then $a(x) \in o(b(x))$ implies

$$\lim_{x \to \infty} \frac{a(x)}{c(x)} = \lim_{x \to \infty} \frac{c(x)}{b(x)} = \lim_{x \to \infty} \sqrt{\frac{a(x)}{b(x)}} = 0$$

as required. $\square$

**Lemma A.2.** *Let $z_1, \ldots, z_n$ be i.i.d. variables in $\{0, 1\}$ and let $Z = \sum_{i=1}^{n} z_i$. If $\mathbb{E}[Z] \geq M$, then $\Pr[Z \leq M/2] \leq \exp(-M/8)$.*

*Proof.* By the Chernoff bound for i.i.d. binary variables, we have $\Pr[Z \leq \mathbb{E}[Z]/2] \leq \exp(-\mathbb{E}[Z]/8)$. Since $-\mathbb{E}[Z] \leq -M$ and $\exp$ is an increasing function, we have $\exp(-\mathbb{E}[Z]/8) \leq \exp(-M/8)$. Also, $M/2 \leq E[Z]/2$ implies $\Pr[Z \leq M/2] \leq \Pr[Z \leq \mathbb{E}[Z]/2]$. Combining these inequalities proves the lemma. $\square$

**Lemma A.3.** *let $A \subseteq [f(T)]$ be any nonempty subset of sections. Then*

$$\Pr\left[\sum_{j \in A} |V_j| \le \frac{T|A|}{4f(T)}\right] \le \exp\left(\frac{-T}{16f(T)}\right)$$

*Proof.* Fix any $j \in [f(T)]$. For each $t \in [T]$, define the random variable $z_t$ by $z_t = 1$ if $t \in V_j$ for some $j \in A$ and 0 otherwise. We have $t \in V_j$ iff $x_t$ falls within a particular interval of length $\frac{1}{2f(T)}$. Since these intervals are disjoint for different $j$'s, we have $z_t = 1$ iff $x_t$ falls within a portion of the input space with total measure $\frac{|A|}{2f(T)}$. Since $x_t$ is uniformly random across $[0, 1]$, we have $\mathbb{E}[z_t] = \frac{|A|}{2f(T)}$. Then $\mathbb{E}[\sum_{t=1}^{T} z_t] = \mathbb{E}[\sum_{j \in A} |V_j|] = \frac{T|A|}{2f(T)}$. Furthermore, since $x_1, \ldots, x_T$ are i.i.d., so are $z_1, \ldots, z_T$. Then by Lemma A.2,

$$\Pr\left[\sum_{j \in A} |V_j| \le \frac{T|A|}{4f(T)}\right] \le \exp\left(\frac{-T|A|}{16f(T)}\right) \le \exp\left(\frac{-T}{16f(T)}\right)$$

with the last step due to $|A| \ge 1$. $\qquad\square$

**Lemma A.4.** *Independently sample $\mathbf{a} \sim U(\{0, 1\}^{f(T)})$ and $\mathbf{x} \sim U^T$.[13] Then with probability at least $1 - \exp\left(\frac{-T}{16f(T)}\right) - \exp\left(-\frac{f(T) - |Q_T|}{16}\right)$,*

$$\prod_{t=1}^{T} \mu_{f, \mathbf{a}}(x_t, y_t) \le \exp\left(-\frac{LT(f(T) - |Q_T|)}{2^7 f(T)^2}\right)$$

*Proof.* **Part 1: setup.** Let $J_{\neg Q} = \{j \in [f(T)] : x_t \notin X_j \ \forall t \in Q_T\}$ be the set of sections that are never queried. Since each query appears in exactly one section (because each input appears in exactly one section), $|J_{\neg Q}| \ge f(T) - |Q_T|$.

For each $j \in J_{\neg Q}$, let $y_j$ be the action taken most frequently among time steps in $V_j$:

$$y_j = \arg\max_{y \in \{0, 1\}} \left|\{t \in V_j : y = y_t\}\right|$$

Let $\bar{J} = \{j \in J_{\neg Q} : a_j \ne y_j\}$. For each $j \in \bar{J}$, let $V_j' = \{t \in V_j : y_t \ne a_j\}$ be the set of time steps where the agent chooses the wrong action (assuming payoff function $\mu_{f, \mathbf{a}}$).

**Part 2: $\bar{J}$ is not too small.** Define a random variable $z_j = \mathbf{1}_{j \in \bar{J}}$ for each $j \in J_{\neg Q}$. By definition, if $j \in J_{\neg Q}$, no input in $X_j$ is queried. Since queries outside of $X_j$ provide no information about $a_j$, the agent's actions must be independent of $a_j$. In particular, the random variables $a_j$ and $y_j$ are independent. Combining that independence with $\Pr[a_j = 0] = \Pr[a_j = 1] = 0.5$ yields $\Pr[z_j = 1] = 0.5$ for all $j \in J_{\neg Q}$. Furthermore, since $a_1, \ldots, a_{f(T)}$ are independent, the random variables $\{z_j : j \in J_{\neg Q}\}$ are also independent. Since $\mathbb{E}[|\bar{J}|] = \mathbb{E}[\sum_{j \in J_{\neg Q}} z_j] = |J_{\neg Q}|/2 \ge \frac{f(T) - |Q_T|}{2}$, Lemma A.2 implies that

$$\Pr\left[|\bar{J}| \le \frac{f(T) - |Q_T|}{4}\right] \le \exp\left(-\frac{f(T) - |Q_T|}{16}\right)$$

**Part 3: $|V_j'| \ge |V_j|/2$.** Since $j \in J_{\neg Q}$, the mentor is not queried on any time step $t \in V_j$, so $y_t \in \{0, 1\}$ for all $t \in V_j$. Since the agent chooses one of two actions for each $t \in V_j$, the more frequent action must be chosen chosen at least half of the time: $y_t = y_j$ for at least half of the time steps in $V_j$. Since $a_j \ne y_j$ for $j \in \bar{J}$, we have $y_t = y_j \ne a_j$ for those time steps, so $|V_j'| \ge |V_j|/2$.

**Part 4: a bound in terms of $\bar{J}$ and $V_j$.** Consider any $j \in \bar{J}$ and $t \in V_j' \subseteq V_j$. By definition of $V_j$, we have $|m_j - x_t| \le \frac{1}{4f(T)}$. Then by definition of $\mu_{f, \mathbf{a}}$,

$$\mu_{f, \mathbf{a}}(x_t, y_t) = 1 - L\left(\frac{1}{2f(T)} - |x_t - m_j|\right)$$

---

[13]That is, the entire set $\{a_1, \ldots, a_{f(T)}, x_1, \ldots, x_T\}$ is mutually independent.

$$\leq 1 - L\left(\frac{1}{2f(T)} - \frac{1}{4f(T)}\right)$$

$$= = 1 - \frac{L}{4f(T)}$$

Now aggregating across time steps,

$$\prod_{t=1}^{T} \mu_{f,\mathbf{a}}(x_t, y_t) \leq \prod_{j \in \bar{J}} \prod_{t \in V_j'} \mu_{f,\mathbf{a}}(x_t, y_t) \qquad (\mu_{f,\mathbf{a}}(x_t, y_t) \in [0,1] \text{ for all } t)$$

$$\leq \prod_{j \in \bar{J}} \left(1 - \frac{L}{4f(T)}\right)^{|V_j'|} \qquad (\text{bound on } \mu_{f,\mathbf{a}}(x_t, y_t) \text{ when } t \in V_j')$$

$$\leq \prod_{j \in \bar{J}} \left(1 - \frac{L}{4f(T)}\right)^{|V_j|/2} \qquad (|V_j'| \geq |V_j|/2)$$

The last step also relies on $1 - \frac{L}{4f(T)} \in [0,1]$, which is due to $L \leq 1$ and $f(T) \geq 1$. Converting into log space and using the standard inequality $\log(1+x) \leq x$ for all $x \in \mathbb{R}$, we have

$$\log \prod_{t=1}^{T} \mu_{f,\mathbf{a}}(x_t, y_t) \leq \log \prod_{j \in \bar{J}} \left(1 - \frac{L}{4f(T)}\right)^{|V_j|/2}$$

$$= \sum_{j \in \bar{J}} \frac{|V_j|}{2} \log\left(1 - \frac{L}{4f(T)}\right)$$

$$\leq -\sum_{j \in \bar{J}} \frac{L|V_j|}{8f(T)}$$

**Part 5: putting it all together.** By Lemma A.3, Part 2 of this lemma, and the union bound, with probability at least $1 - \exp\left(\frac{-T}{16f(T)}\right) - \exp\left(-\frac{f(T)-|Q_T|}{16}\right)$ we have $\sum_{j \in \bar{J}} |V_j| \geq \frac{T|\bar{J}|}{4f(T)}$ for all $j \in [f(T)]$ and $|\bar{J}| \geq \frac{f(T)-|Q_T|}{4}$. Assuming those inequalities hold, we have

$$\log \prod_{t=1}^{T} \mu_{f,\mathbf{a}}(x_t, y_t) \leq -\sum_{j \in \bar{J}} \frac{L|V_j|}{8f(T)}$$

$$\leq -\frac{L}{8f(T)} \cdot \frac{T|\bar{J}|}{4f(T)}$$

$$\leq -\frac{L}{8f(T)} \cdot \frac{T}{4f(T)} \cdot \frac{f(T)-|Q_T|}{4}$$

$$= -\frac{LT(f(T)-|Q_T|)}{2^7 f(T)^2}$$

Exponentiating both sides proves the lemma. $\qquad\square$

Let $\alpha(T) = \exp\left(\frac{-T}{16f(T)}\right) + \exp\left(-\frac{f(T)-|Q_T|}{16}\right)$ for brevity.

**Theorem 4.1.** *The worst-case expected regret of any algorithm with sublinear queries goes to 1 as $T$ goes to infinity. Formally,* $\lim_{T \to \infty} \sup_{\mu, \pi^m} \mathbb{E}[R_T] = 1$.

*Proof.* If the algorithm has sublinear queries, then there exists $g(T) \in o(T)$ such that $|Q_T| \leq g(T)$ always. Let $f(T) = \sqrt{(g(T)+1)T}$. Then by Lemma A.1, $g(T) \in o(f(T))$ and $f(T) \in o(T)$. Combining this with $|Q_T| \leq g(T)$, we get $\lim_{T \to \infty} \alpha(T) = 0$. Also, since $g(T) \in o(f(T))$, there

exists $T_0$ such that $|Q_T| \leq g(T) \leq f(T)/2$ for all $T \geq T_0$. Combining this with Lemma A.4 and noting that $\prod_{t=1}^{T} \mu_{f,\mathbf{a}}(x_t, y_t) \leq 1$, we have

$$\mathop{\mathbb{E}}_{\mathbf{a} \sim U(\{0,1\}^{f(T)})} \mathop{\mathbb{E}}_{\mathbf{x} \sim U^T, \mathbf{y}} \prod_{t=1}^{T} \mu_{f,\mathbf{a}}(x_t, y_t)$$

$$\leq \alpha(T) \cdot 1 + \big(1 - \alpha(T)\big) \exp\left(-\frac{LT(f(T) - |Q_T|)}{2^7 f(T)^2}\right)$$

$$\leq \alpha(T) + \big(1 - \alpha(T)\big) \exp\left(-\frac{LT f(T)/2}{2^7 f(T)^2}\right)$$

$$= \alpha(T) + \big(1 - \alpha(T)\big) \exp\left(-\frac{LT}{2^8 f(T)}\right)$$

whenever $T \geq T_0$. Since $\prod_{t=1}^{T} \mu_{f,\mathbf{a}}^m(x_t) = 1$ always, we have[14]

$$\sup_{\mu} \mathop{\mathbb{E}}_{\mathbf{x} \sim U^T, \mathbf{y}} R_T(\mathbf{x}, \mathbf{y}, \mu, \pi^m) \geq \mathop{\mathbb{E}}_{\mathbf{a} \sim U(\{0,1\}^{f(T)})} \mathop{\mathbb{E}}_{\mathbf{x} \sim U^T, \mathbf{y}} R_T(\mathbf{x}, \mathbf{y}, \mu_{f,\mathbf{a}}, \pi^m)$$

$$= \mathop{\mathbb{E}}_{\mathbf{a} \sim U(\{0,1\}^{f(T)})} \mathop{\mathbb{E}}_{\mathbf{x} \sim U^T, \mathbf{y}} \left[\prod_{t=1}^{T} \mu_{f,\mathbf{a}}^m(x_t) - \prod_{t=1}^{T} \mu_{f,\mathbf{a}}(x_t, y_t)\right]$$

$$\geq 1 - \alpha(T) - \big(1 - \alpha(T)\big) \exp\left(-\frac{LT}{2^8 f(T)}\right)$$

Therefore

$$\lim_{T \to \infty} \sup_{\mu} \mathop{\mathbb{E}}_{\mathbf{x} \sim U^T, \mathbf{y}} R_T(\mathbf{x}, \mathbf{y}, \mu, \pi^m) \geq 1 - \lim_{T \to \infty} \alpha(T) - \big(1 - \lim_{T \to \infty} \alpha(T)\big) \cdot \exp\left(\lim_{T \to \infty} -\frac{LT}{2^8 f(T)}\right)$$

$$= 1 - 0 - (1 - 0) \cdot \exp(-\infty)$$

$$= 1$$

as required. $\qquad\square$

## B  PROOF OF THEOREM 5.2

### B.1  CONTEXT ON LEMMA 5.4

Before diving into the main proof, we provide some context on Lemma 5.4 from Section 5:

**Lemma 5.4** (Lemma 3.5 in Russo et al. (2024)). *Assume $\tilde{\Pi}$ is finite. Then for any loss function $\ell : [T] \times \tilde{\Pi} \to [0, 1]$ and query probability $p$,* HEDGEWITHQUERIES *enjoys the regret bound*

$$\sum_{t=1}^{T} \mathbb{E}[\ell(t, \pi_t)] - \min_{\tilde{\pi} \in \tilde{\Pi}} \sum_{t=1}^{T} \ell(t, \pi) \leq \frac{2 \log |\tilde{\Pi}|}{p^2}$$

*where $\pi_t$ is the policy chosen at time $t$ and the expectation is over the randomness of the algorithm.*

Lemma 5.4 is a restatement and simplification of Lemma 3.5 in Russo et al. (2024). First, Russo et al. (2024) parametrize their algorithm by the expected number of queries $\hat{k}$ instead of the query probability $p = \hat{k}/T$. Second, Russo et al. (2024) include a second parameter $k$, which is the eventual target number of queries for their unconditional query bound. In our case, an expected query bound is sufficient, so we simply set $k = \hat{k}$. Third, Russo et al. (2024) provide a second bound which is tighter for small $k$; that bound is less useful for us so we omit it. Fourth, their number of actions $n$ is equal to $|\tilde{\Pi}|$ in our setting. (Their actions correspond to policies in $\tilde{\Pi}$, not our actions in $\mathcal{Y}$.) Since Russo et al. (2024) set $\eta = \max\left(\frac{1}{T}\sqrt{\frac{\hat{k} \log n}{2}}, \frac{k\hat{k}}{\sqrt{2}T^2}\right)$, we end up with $\eta = \max\left(\sqrt{\frac{p \log |\tilde{\Pi}|}{2T}}, \frac{p^2}{\sqrt{2}}\right)$. Algorithm 2 provides precise pseudocode for the HEDGEWITHQUERIES algorithm to which Lemma 5.4 refers.

---

[14]Fubini's theorem means we need not worry about the order of the expectation operators.

---

**Algorithm 2** A variant of the Hedge algorithm which only observes losses in response to queries.

---

1: **function** HEDGEWITHQUERIES($p \in (0, 1]$, finite policy class $\tilde{\Pi}$, unknown $\ell : [T] \times \tilde{\Pi} \to [0, 1]$)
2:     $w(\pi) \leftarrow 1$ for all $\pi \in \tilde{\Pi}$
3:     $\eta \leftarrow \max \left( \sqrt{\frac{p \log |\tilde{\Pi}|}{2T}}, \frac{p^2}{\sqrt{2}} \right)$
4:     **for** $t$ from 1 **to** $T$ **do**
5:         hedgeQuery $\leftarrow$ true with probability $p$ else false
6:         **if** hedgeQuery **then**
7:             Query and observe $\ell(t, \pi)$ for all $\pi \in \tilde{\Pi}$
8:             $\ell^* \leftarrow \min_{\pi \in \tilde{\Pi}} \ell(t, \pi)$
9:             $w(\pi) \leftarrow w(\pi) \cdot \exp(-\eta(\ell(t, \pi) - \ell^*))$ for all $\pi \in \tilde{\Pi}$
10:            Select policy $\arg \min_{\pi \in \tilde{\Pi}} \ell(t, \pi)$
11:        **else**
12:            $P(\pi) \leftarrow w(\pi) / \sum_{\pi' \in \tilde{\Pi}} w(\pi')$ for all $\pi \in \tilde{\Pi}$
13:            Sample $\pi_t \sim P$
14:            Select policy $\pi_t$

---

## B.2   MAIN PROOF

We use the following notation throughout the proof:

1. For each $t \in [T]$, let $X_t$ refer to the value of $X$ at the start of time step $t$.

2. Let $V_T = \{t \in [T] : \pi_t(x_t) \neq \pi^m(x_t)\}$ be the set of time steps where Hedge's proposed action doesn't match the mentor's. Note that $|V_T|$ upper bounds the number of mistakes the algorithm makes (the number of mistakes could be smaller, since the algorithm sometimes queries instead of taking action $\pi_t(x_t)$).

3. For $X \subseteq \mathcal{X}$, let $\mathrm{vol}(X)$ denote the $n$-dimensional Lebesgue measure of $X$.

4. With slight abuse of notation, we will use inequalities of the form $f(T) \leq g(T) + O(h(T))$ to mean that there exists a constant $C$ such that $f(T) \leq g(T) + Ch(T)$.

5. We will use "Case 1" to refer to finite VC dimension and $\sigma$-smooth $\mathbf{x}$ and "Case 2" to refer to finite Littlestone dimension. In Case 1, expectations are over the randomness of both $\mathbf{x}$ and the algorithm, while in Case 2, expectations are over just the randomness of the algorithm. When we need to distinguish, we use $\mathbb{E}_{\mathbf{y}}$ to denote the expectation over randomness of the algorithm and $\mathbb{E}_{\mathbf{x} \sim \mathcal{D}}$ to denote the expectation over $\mathbf{x}$.

**Lemma B.1.** *Under the conditions of Theorem 5.1, Algorithm 1 satisfies*

$$\mathbb{E}[|V_T|] \in O \left( \frac{d}{\sigma} T \varepsilon \log(1/\varepsilon) \log T \right)$$

*Proof.* Define $\ell : [T] \times \tilde{\Pi} \to [0, 1]$ by $\ell(t, \pi) = \mathbf{1}(\pi(x_t) \neq \pi^m(x_t))$, and let $w^h$ and $\pi_t^h$ denote the values of $w$ and $\pi_t$ respectively in HEDGEWITHQUERIES, while $w$ and $\pi_t$ refer to the variables in Algorithm 1. Then $w$ and $w^h$ evolve in the exact same way, so the distributions of $\pi_t$ and $\pi_t^h$ coincide. Thus by Lemma 5.4,

$$\mathbb{E}_{\mathbf{y}} \left[ \sum_{t=1}^T \ell(t, \pi_t) \right] - \min_{\tilde{\pi} \in \tilde{\Pi}} \sum_{t=1}^T \ell(t, \tilde{\pi}) \leq \frac{2 \log |\tilde{\Pi}|}{p^2} = 2T \varepsilon \log |\tilde{\Pi}|$$

Since Lemma 5.4 holds for any loss function, the bound above holds for any $\mathbf{x} \in S^T$, so the bound also holds in expectation over $\mathbf{x} \sim \mathcal{D}$ (which is needed for Case 1). Next, observe that $|V_T| = \sum_{t=1}^T \mathbf{1}(\pi_t(x_t) \neq \pi^m(x_t)) = \sum_{t=1}^T \ell(t, \pi_t)$, so

$$\mathbb{E}_{\mathbf{y}}[|V_T|] \leq 2T \varepsilon \log |\tilde{\Pi}| + \min_{\tilde{\pi} \in \tilde{\Pi}} \sum_{t=1}^T \mathbf{1}(\tilde{\pi}(x_t) \neq \pi^m(x_t))$$

**Case 1:** Since $\tilde{\Pi}$ is a smooth $\varepsilon$-cover of $\Pi$, we have

$$\mathbb{E}_{\mathbf{x} \sim \mathcal{D}} \left[ \min_{\tilde{\pi} \in \tilde{\Pi}} \sum_{t=1}^{T} \mathbf{1}(\tilde{\pi}(x_t) \neq \pi^m(x_t)) \right] \leq \mathbb{E}_{\mathbf{x} \sim \mathcal{D}} \left[ \sup_{\pi \in \Pi} \min_{\tilde{\pi} \in \tilde{\Pi}} \sum_{t=1}^{T} \mathbf{1}(\tilde{\pi}(x_t) \neq \pi(x_t)) \right]$$

$$\in O\left( \frac{1}{\sigma} T \varepsilon \log T \sqrt{d \log(1/\varepsilon)} \right)$$

with the first step due to $\pi^m \in \Pi$ and the second step due to Lemma 5.3. The last component we need is that $|\tilde{\Pi}| \leq (41/\varepsilon)^d$ by construction (and such a $\tilde{\Pi}$ is guaranteed to exist by Lemma 5.1). Combining the above inequalities and taking the expectation over $\mathbf{x} \sim \mathcal{D}$ (in addition to the randomness of the algorithm), we get

$$\mathbb{E}_{\mathbf{x} \sim \mathcal{D}, \mathbf{y}} [|V_T|] \leq 2T\varepsilon \log |\tilde{\Pi}| + \mathbb{E}_{\mathbf{x} \sim \mathcal{D}} \left[ \min_{\tilde{\pi} \in \tilde{\Pi}} \sum_{t=1}^{T} \mathbf{1}(\tilde{\pi}(x_t) \neq \pi^m(x_t)) \right]$$

$$\leq 2dT\varepsilon \log(41/\varepsilon) + O\left( \frac{1}{\sigma} T \varepsilon \log T \sqrt{d \log(1/\varepsilon)} \right)$$

$$\in O\left( \frac{d}{\sigma} T \varepsilon \log(1/\varepsilon) \log T \right)$$

**Case 2:** Since $\tilde{\Pi}$ is an adversarial cover of $\Pi$ and $\pi^m \in \Pi$, there exists $\tilde{\pi} \in \tilde{\Pi}$ such that $\sum_{t=1}^{T} \mathbf{1}(\tilde{\pi}(x_t) \neq \pi^m(x_t)) = 0$. Since $|\tilde{\Pi}| \leq (eT/d)^d$ (with such a $\tilde{\Pi}$ guaranteed to exist by Lemma 5.2),

$$\mathbb{E}_{\mathbf{y}}[|V_T|] \leq 2T\varepsilon \log |\tilde{\Pi}| + \min_{\tilde{\pi} \in \tilde{\Pi}} \sum_{t=1}^{T} \mathbf{1}(\tilde{\pi}(x_t) \neq \pi^m(x_t))$$

$$\leq 2T\varepsilon d \ln(eT/d)$$

$$\in O\left( \frac{d}{\sigma} T \varepsilon \log(1/\varepsilon) \log T \right)$$

as required. $\qquad \square$

**Lemma B.2.** *For all $t \in [T]$, $\mu(x_t, y_t) \geq \mu^m(x_t) - L\varepsilon^{1/n}$.*

*Proof.* Fix any $t \in [T]$. If $t \in Q_T$, then $\mu(x_t, y_t) = \mu^m(x_t)$, so assume $t \notin Q_T$. Let $(x', y') = \arg \min_{(x,y) \in X_t : \pi_t(x_t)=y} ||x_t - x||$. Since $t \notin Q_T$, we must have $||x_t - x'|| \leq \varepsilon^{1/n}$.

We have $y' = \pi^m(x')$ by construction of $X_t$ and $\pi_t(x_t) = y'$ by construction of $y'$. Combining these with the local generalization assumption, we get

$$\mu(x_t, y_t) = \mu(x_t, \pi_t(x_t)) = \mu(x_t, \pi^m(x')) \geq \mu^m(x_t) - L||x_t - x'|| \geq \mu^m(x_t) - L\varepsilon^{1/n}$$

as required. $\qquad \square$

**Lemma B.3.** *Assume $a_1, \ldots, a_T, b_1, \ldots, b_T \in [0, 1]$ and $a_t \geq b_t$ for all $t \in [T]$. Then*

$$\prod_{t=1}^{T} a_t - \prod_{t=1}^{T} b_t \leq \sum_{t=1}^{T} a_t - \sum_{t=1}^{T} b_t$$

*Proof.* We proceed by induction on $T$. The claim is trivially satisfied for $T = 1$, so suppose $T > 1$ and assume that $\prod_{t=1}^{T-1} a_t - \prod_{t=1}^{T-1} b_t \leq \sum_{t=1}^{T-1} a_t - \sum_{t=1}^{T-1} b_t$. Then

$$\sum_{t=1}^{T} a_t - \sum_{t=1}^{T} b_t - \prod_{t=1}^{T} a_t + \prod_{t=1}^{T} b_t = a_T \sum_{t=1}^{T-1} a_t - b_T \sum_{t=1}^{T-1} b_t - a_T \prod_{t=1}^{T-1} a_t + b_T \prod_{t=1}^{T-1} b_t$$

$$= a_T \left( \sum_{t=1}^{T-1} a_t - \prod_{t=1}^{T-1} a_t \right) - b_T \left( \sum_{t=1}^{T-1} b_t - \prod_{t=1}^{T-1} b_t \right)$$

Since $T > 1$ and $a_t \in [0, 1]$ for all $t \in [T]$, we have $\sum_{t=1}^{T-1} a_t \geq a_1 \geq \prod_{t=1}^{T-1} a_t$. Thus $\sum_{t=1}^{T-1} a_t - \prod_{t=1}^{T-1} a_t \geq 0$. Combining this with $a_T \geq b_T$, we get

$$
\begin{aligned}
\sum_{t=1}^{T} a_t - \sum_{t=1}^{T} b_t - \prod_{t=1}^{T} a_t + \prod_{t=1}^{T} b_t &= a_T \left( \sum_{t=1}^{T-1} a_t - \prod_{t=1}^{T-1} a_t \right) - b_T \left( \sum_{t=1}^{T-1} b_t - \prod_{t=1}^{T-1} b_t \right) \\
&\geq b_T \left( \sum_{t=1}^{T-1} a_t - \prod_{t=1}^{T-1} a_t \right) - b_T \left( \sum_{t=1}^{T-1} b_t - \prod_{t=1}^{T-1} b_t \right) \\
&= b_T \left( \sum_{t=1}^{T-1} a_t - \prod_{t=1}^{T-1} a_t - \sum_{t=1}^{T-1} b_t + \prod_{t=1}^{T-1} b_t \right) \\
&\geq 0
\end{aligned}
$$

The last step is due to $b_T \geq 0$ and our assumption of $\prod_{t=1}^{T-1} a_t - \prod_{t=1}^{T-1} b_t \leq \sum_{t=1}^{T-1} a_t - \sum_{t=1}^{T-1} b_t$. □

**Lemma B.4.** *Under the conditions of Theorem 5.1, Algorithm 1 satisfies*

$$
\mathbb{E}\left[R_T\right] \in O\left( \frac{dL}{\sigma} T \varepsilon^{1+1/n} \log(1/\varepsilon) \log T \right)
$$

$$
\mathbb{E}\left[ \sum_{t=1}^{T} \mu^m(x_t) - \sum_{t=1}^{T} \mu(x_t, y_t) \right] \in O\left( \frac{dL}{\sigma} T \varepsilon^{1+1/n} \log(1/\varepsilon) \log T \right)
$$

*Proof.* We first claim that $y_t = \pi^m(x_t)$ for all $t \notin V_T$. If $t \in Q_T$, the claim is immediate; if not, we have $y_t = \pi_t(x_t)$, and $\pi_t(x_t) = \pi^m(x_t)$ due to $t \notin V_T$. Thus $\min(\mu^m(x_t), \mu(x_t, y_t)) = \mu^m(x_t)$ for $t \notin V_T$.

We next claim that $\mu^m(x_t) - \min(\mu^m(x_t), \mu(x_t, y_t)) \leq L\varepsilon^{1/n}$ for all $t \in [T]$. If $\mu(x_t, y_t) \leq \mu^m(x_t)$, this follows from Lemma B.2. If $\mu(x_t, y_t) > \mu^m(x_t)$, then $\mu^m(x_t) - \min(\mu^m(x_t), \mu(x_t, y_t)) = 0 \leq L\varepsilon^{1/n}$. Therefore

$$
\begin{aligned}
\sum_{t=1}^{T} \left( \mu^m(x_t) - \min(\mu^m(x_t), \mu(x_t, y_t)) \right) &\leq \sum_{t \in V_T} \left( \mu^m(x_t) - \min(\mu^m(x_t), \mu(x_t, y_t)) \right) \\
&\leq \sum_{t \in V_T} L\varepsilon^{1/n} \\
&= |V_T| L\varepsilon^{1/n}
\end{aligned}
$$

Now let $a_t = \mu^m(x_t)$ and $b_t = \min(\mu^m(x_t), \mu(x_t, y_t))$ for all $t \in [T]$. Then by Lemma B.3,

$$
\prod_{t=1}^{T} \mu^m(x_t) - \prod_{t=1}^{T} \min(\mu^m(x_t), \mu(x_t, y_t)) \leq \sum_{t=1}^{T} \left( \mu^m(x_t) - \min(\mu^m(x_t), \mu(x_t, y_t)) \right)
$$

Since $\mu(x_t, y_t) \geq \min(\mu^m(x_t), \mu(x_t, y_t))$ for all $t \in [T]$, we have

$$
\begin{aligned}
R_T &= \prod_{t=1}^{T} \mu^m(x_t) - \prod_{t=1}^{T} \mu(x_t, y_t) \\
&\leq \sum_{t=1}^{T} \left( \mu^m(x_t) - \min(\mu^m(x_t), \mu(x_t, y_t)) \right) \\
&\leq |V_T| L\varepsilon^{1/n}
\end{aligned}
$$

Since we also have $\sum_{t=1}^{T} \mu^m(x_t) - \sum_{t=1}^{T} \mu(x_t, y_t) \leq \sum_{t=1}^{T} (\mu^m(x_t) - \min(\mu^m(x_t), \mu(x_t, y_t)))$,

$$
\mathbb{E}[R_T] \leq L\varepsilon^{1/n} \mathbb{E}\left[ |V_T| \right]
$$

$$\mathbb{E}\left[\sum_{t=1}^{T}\mu^m(x_t) - \sum_{t=1}^{T}\mu(x_t, y_t)\right] \le L\varepsilon^{1/n}\,\mathbb{E}\left[|V_T|\right]$$

Applying Lemma B.1 completes the proof. $\qquad\square$

**Definition B.1.** Let $(K, ||\cdot||)$ be a normed vector space and let $\delta > 0$. Then $X \subseteq K$ is a $\delta$-packing of $K$ if for all $x, y \in X$, $||x - y|| > \delta$. The $\delta$-packing number of $K$, denoted $\mathcal{M}(K, ||\cdot||, \delta)$, is the maximum cardinality of any $\delta$-packing of $K$.

In this paper, we only consider the Euclidean distance norm, so we just write $M(K, ||\cdot||, \delta) = M(K, \delta)$.

**Lemma B.5** (Theorem 14.2 in Wu (2020)). *If $K \subset \mathbb{R}^n$ is convex, bounded, and contains a ball with radius $\delta > 0$, then*

$$\mathcal{M}(K, \delta) \le \frac{3^n \operatorname{vol}(K)}{\delta^n \operatorname{vol}(B)}$$

*where $B$ is a unit ball.*

**Lemma B.6** (Jung's Theorem (Jung, 1901)). *If $X \subset \mathbb{R}^n$ is compact, then there exists a closed ball with radius at most $\operatorname{diam}(X)\sqrt{\dfrac{n}{2(n+1)}}$ containing $X$.*

**Lemma B.7.** *Under the conditions of Theorem 5.1, Algorithm 1 satisfies*

$$\mathbb{E}[|Q_T|] \in O\left(\sqrt{\frac{T}{\varepsilon}} + \frac{d}{\sigma}T\varepsilon\log(1/\varepsilon)\log T + \frac{\operatorname{diam}(\mathbf{x})^n}{\varepsilon}\right)$$

*Proof.* If $t \in Q_T$, then either $\texttt{hedgeQuery} = \texttt{true}$ or $\min_{(x,y)\in X_t:\pi_t(x_t)=y}||x_t - x|| > r$. The expected number of time steps with $\texttt{hedgeQuery} = \texttt{true}$ is $pT = \sqrt{T/\varepsilon}$, so let $\hat{X} = \{x_t : t \in Q_T \text{ and } \min_{(x,y)\in X_t:\pi_t(x_t)=y}||x_t - x|| > r)\}$. We further subdivide $\hat{X}$ into $\hat{X}_1 = \{x_t \in \hat{X} : \pi_t(x_t) \neq \pi^m(x_t)\}$ and $\hat{X}_2 = \{x_t \in \hat{X} : \pi_t(x_t) = \pi^m(x_t)\}$. Since $\hat{X}_1 \subseteq V_T$, Lemma B.1 implies that $\mathbb{E}[|\hat{X}_1|] \in O\left(\frac{d}{\sigma}T\varepsilon\log(1/\varepsilon)\log T\right)$.

Next, fix an $y \in \mathcal{Y}$ and let $X_y = \{x \in \mathbf{x} : \pi^m(x) = y\}$ be the set of observed inputs which share a mentor action. We claim that $\hat{X}_2 \cap X_y$ is a packing of $X_y$. Suppose instead that there exists $x, x' \in \hat{X}_2 \cap X_y$, with $||x - x'|| \le \varepsilon^{1/n}$. WLOG assume $x$ was queried after $x'$ and let $t$ be the time step on which $x$ was queried. Then $(x', \pi^m(x')) \in X_t$. Also, $x, x' \in \hat{X}_2 \cap X_y$ implies that and $\pi_t(x_t) = \pi^m(x_t) = y = \pi^m(x')$. Therefore

$$\min_{(x'',y'')\in X_t:y''=\pi_t(x_t)}||x_t - x''|| \le ||x_t - x'|| \le \varepsilon^{1/n}$$

which contradicts $x_t \in \hat{X}$. Thus $\hat{X}_2 \cap X_y$ is a $\varepsilon^{1/n}$-packing of $X_y$.

By Lemma B.6, there exists a ball $B_1$ of diameter $\operatorname{diam}(\mathbf{x})\sqrt{\frac{n}{2(n+1)}}$ which contains $\mathbf{x}$. Let $R = \operatorname{diam}(\mathbf{x})\sqrt{\frac{n}{8(n+1)}}$ denote the radius of $B_1$. Let $B_2$ be the ball with the same center as $B_1$ but with radius $\max(R, \varepsilon^{1/n})$. Since $X_y \subset \mathbf{x} \subset B_1 \subset B_2$, $\hat{X}_2 \cap X_y$ is also a $\varepsilon^{1/n}$-packing of $B_2$. Also, $B_2$ must contain a ball of radius $\varepsilon^{1/n}$, so Lemma B.5 implies that

$$
\begin{aligned}
|\hat{X}_2 \cap X_y| &\le \mathcal{M}(B_2, \varepsilon^{1/n}) \\
&\le \frac{3^n \operatorname{vol}(B_2)}{\varepsilon \operatorname{vol}(B)} \\
&= \left(\max(R, \varepsilon^{1/n})\right)^n \frac{3^n \operatorname{vol}(B)}{\varepsilon \operatorname{vol}(B)} \\
&= \max\left(\operatorname{diam}(\mathbf{x})^n\left(\frac{n}{8(n+1)}\right)^{n/2}, \varepsilon\right)\frac{3^n}{\varepsilon}
\end{aligned}
$$

$$\leq O\left(\frac{\mathrm{diam}(\mathbf{x})^n}{\varepsilon} + 1\right)$$

(The $+1$ is necessary for now since $\mathrm{diam}(\mathbf{x})$ could theoretically be zero.) Therefore

$$\mathbb{E}[|Q_T|] = \sqrt{\frac{T}{\varepsilon}} + \mathbb{E}[|\hat{X}|]$$

$$= \sqrt{\frac{T}{\varepsilon}} + \mathbb{E}[|\hat{X}_1|] + \mathbb{E}\left[\sum_{y\in\mathcal{Y}} |\hat{X}_2 \cap X_y|\right]$$

$$\leq \sqrt{\frac{T}{\varepsilon}} + O\left(\frac{d}{\sigma}T\varepsilon\log(1/\varepsilon)\log T\right) + \sum_{y\in\mathcal{Y}} O\left(\frac{\mathrm{diam}(\mathbf{x})^n}{\varepsilon} + 1\right)$$

$$\leq \sqrt{\frac{T}{\varepsilon}} + O\left(\frac{d}{\sigma}T\varepsilon\log(1/\varepsilon)\log T\right) + |\mathcal{Y}| \cdot O\left(\frac{\mathrm{diam}(\mathbf{x})^n}{\varepsilon} + 1\right)$$

$$\leq O\left(\sqrt{\frac{T}{\varepsilon}} + \frac{d}{\sigma}T\varepsilon\log(1/\varepsilon)\log T + \frac{\mathrm{diam}(\mathbf{x})^n}{\varepsilon}\right)$$

as required. $\qquad\square$

Theorem 5.1 follows from Lemmas B.4 and B.7:

**Theorem 5.1.** *Let $\mathcal{Y} = \{0,1\}$. Assume $\pi^m \in \Pi$ where either (1) $\Pi$ has finite VC dimension $d$, $\mathbf{x}$ is $\sigma$-smooth, and $\varepsilon T \log T > 12\sigma d\log(4e^2/\varepsilon)$ or (2) $\Pi$ has finite Littlestone dimension $d$. Then for any $T \in \mathbb{N}$ and $\varepsilon > 0$, Algorithm 1 satisfies*

$$\mathbb{E}[R_T] \in O\left(\frac{dL}{\sigma}T\varepsilon^{1+1/n}\log(1/\varepsilon)\log T\right)$$

$$\mathbb{E}[|Q_T|] \in O\left(\sqrt{\frac{T}{\varepsilon}} + \frac{d}{\sigma}T\varepsilon\log(1/\varepsilon)\log T + \frac{\mathrm{diam}(\mathbf{x})^n}{\varepsilon}\right)$$

We then perform some arithmetic to get Theorem 5.2:

**Theorem 5.2.** *Let $\mathcal{Y} = \{0,1\}$. Assume $\pi^m \in \Pi$ where either (1) $\Pi$ has finite VC dimension $d$ and $\mathbf{x}$ is $\sigma$-smooth or (2) $\Pi$ has finite Littlestone dimension $d$. Then for any $T \in \mathbb{N}$, Algorithm 1 with $\varepsilon = T^{\frac{-2n}{2n+1}}$ satisfies*

$$\mathbb{E}[R_T] \in O\left(\frac{dL}{\sigma}T^{\frac{-1}{2n+1}}\log T\right)$$

$$\mathbb{E}[|Q_T|] \in O\left(T^{\frac{4n+1}{4n+2}}\left(\frac{d}{\sigma}\log T + \mathrm{diam}(\mathbf{x})^n\right)\right)$$

*Proof.* We have

$$\mathbb{E}[R_T] \in O\left(\frac{dL}{\sigma}T^{1-\frac{2n}{2n+1}-\frac{2}{2n+1}}\log(1/\varepsilon)\log T\right)$$

$$= O\left(\frac{dL}{\sigma}T^{\frac{-1}{2n+1}}\log T\right)$$

and

$$\mathbb{E}[|Q_T|] \in O\left(\sqrt{T^{1+\frac{2n}{2n+1}}} + \frac{d}{\sigma}T^{1-\frac{-2n}{2n+1}}\log(T^{\frac{2n}{2n+1}})\log T + T^{\frac{2n}{2n+1}}\mathrm{diam}(\mathbf{x})^n\right)$$

$$= O\left(T^{\frac{2n+0.5}{2n+1}} + \frac{d}{\sigma}T^{\frac{1}{2n+1}}\log T + T^{\frac{2n}{2n+1}}\mathrm{diam}(\mathbf{x})^n\right)$$

$$\leq O\left(T^{\frac{4n+1}{4n+2}}\left(\frac{d}{\sigma}\log T + \mathrm{diam}(\mathbf{x})^n\right)\right)$$

$$\square$$

If we instead use the second bound from Lemma B.4, the same arithmetic gives us:

**Theorem 5.3.** *Let $\mathcal{Y} = \{0, 1\}$. Assume $\pi^m \in \Pi$ where either (1) $\Pi$ has finite VC dimension $d$ and $\mathbf{x}$ is $\sigma$-smooth or (2) $\Pi$ has finite Littlestone dimension $d$. Then for any $T \in \mathbb{N}$, Algorithm 1 with $\varepsilon = T^{\frac{-2n}{2n+1}}$ satisfies*

$$\mathbb{E}\left[\sum_{t=1}^{T} \mu^m(x_t) - \sum_{t=1}^{T} \mu(x_t, y_t)\right] \in O\left(\frac{dL}{\sigma} T^{\frac{-1}{2n+1}} \log T\right)$$

### B.3 ADAPTIVE ADVERSARIES

If $s_t$ is allowed to depend on the events of prior time steps, we say that the adversary is adaptive. In contrast, a non-adaptive or "oblivious" adversary must choose the entire input upfront. This distinction is not relevant for deterministic algorithms, since an adversary knows exactly how the algorithm will behave for any input. In other words, the adversary gains no new information during the execution of the algorithm. For randomized algorithms, an adaptive adversary can base the choice of $s_t$ on the results of randomization on previous time steps (but not on the current time step), while an oblivious adversary cannot.

In the standard online learning model, Hedge guarantees sublinear regret against both oblivious and adaptive adversaries (Chapter 5 of Slivkins et al. (2019) or Chapter 21 of Shalev-Shwartz & Ben-David (2014)). However, Russo et al. (2024) state their result only for oblivious adversaries. In order for our overall proof of Theorem 5.1 to hold for adaptive adversaries, Lemma 5.4 (Lemma 3.5 in Russo et al. (2024)) must also hold for adaptive adversaries. In this section, we argue why the proof of Lemma 5.4 (Lemma 3.5 in their paper) goes through for adaptive adversaries as well. For this rest of Appendix B.3, lemma numbers refer to the numbering in Russo et al. (2024).

**The importance of independent queries.** Recall from Appendix B.1 that Russo et al. (2024) allow two separate parameters $k$ and $\hat{k}$, which we unify for simplicity. Recall also that Lemma 3.5 refers to the variant of Hedge which queries with probability $p = \hat{k}/T = k/T$ independently on each time step (Algorithm 2. More precisely, on each time step $t$, the algorithm samples a Bernoulli random variable $X_t \sim \text{Ber}(p)$ and queries if $X_t = 1$. The key idea is that $X_t$ is independent of events on previous time steps. Thus even conditioning on the history up to time $t$, for any for any random variable $Y_t$ we can write

$$\mathbb{E}[Y_t] = (1-p)\,\mathbb{E}[Y_t \mid X_t = 0] + p\,\mathbb{E}[Y_t \mid X_t = 1]$$

This insight immediately extends Observation 3.3 to adaptive adversaries (with the minor modification that queries are now issued independently with probability $p$ on each time step instead of issuing $k$ uniformly distributed queries). Specifically, using the notation from Russo et al. (2024) where $i_t$ is the action chosen at time $t$, $i_t^0$ is the action chosen at time $t$ if a query is not issued, and $i_t^*$ is the optimal action at time $t$, we have

$$\mathbb{E}[\ell_t(i_t)] = (1-p)\,\mathbb{E}[\ell_t(i_t^0)] + p\,\mathbb{E}[\ell_t(i_t^*)] = \left(1 - \frac{k}{T}\right)\mathbb{E}[\ell_t(i_t^0)] + \frac{k}{T}\,\mathbb{E}[\ell_t(i_t^*)]$$

The same logic applies to other statements like $\mathbb{E}[\hat{\ell}_t(i) \mid X_{\leq t-1}, I_{\leq t-1}] = \ell_t(i) - \ell_t(i_t^*)$ and immediately extends those statements to adaptive adversaries as well.

**Applying Observation 3.3.** The other tricky part of the proof is applying Observation 3.3 using a new loss function $\hat{\ell}$ defined by $\hat{\ell}_t = \frac{T}{k}(\ell_t(i) - \ell_t(i_t^*))\mathbf{1}(X_t = 1)$. To do so, we must argue that standard Hedge run on $\hat{\ell}$ is the "counterpart without queries" of HEDGEWITHQUERIES. Specifically, both algorithms must have the same weight vectors on every time step, and the only difference should be that HEDGEWITHQUERIES takes the optimal action on each time step independently with probability $p$ (and otherwise behaves the same as standard Hedge). On time steps with $X_t = 0$, standard Hedge observes $\hat{\ell}_t(i) = 0$ for all actions $i$ and thus makes no updates, and HEDGEWITHQUERIES makes no updates by definition. On time steps with $X_t = 1$, both algorithms perform the typical updates $w_{t+1}(i) = w_t(i) \cdot \exp(-\eta(\hat{\ell}_t(i) - \hat{\ell}_t(i_t^*)))$. Thus the weight vectors are the same for both algorithms on every time step. Furthermore, HEDGEWITHQUERIES takes the optimal action at time $t$ iff $X_t = 1$,

---

**Algorithm 3** extends Algorithm 1 to many actions.

---

1: **function** AVOIDCATASTROPHEMANYACTIONS($T \in \mathbb{N}$, $\varepsilon \in \mathbb{R}_{>0}$, $d \in \mathbb{N}$, policy class $\Pi$)
2:     **for** $y \in \mathcal{Y}$ **do**
3:         **if** $\Pi$ has VC dimension $d$ **then**
4:             $\tilde{\Pi}_y \leftarrow$ any smooth $\varepsilon$-cover of $\Pi$ of size at most $(41/\varepsilon)^d$
5:         **else if** $\Pi$ has Littlestone dimension $d$ **then**
6:             $\tilde{\Pi}_y \leftarrow$ any adversarial $\varepsilon$-cover of size at most $(eT/d)^d$
7:     **for** $t$ **from** 1 **to** $T$ **do**
8:         **for** $y \in \mathcal{Y}$ **do**
9:             $b_t^y \leftarrow$ action from running one step of Algorithm 1 on $\Pi_y$ (with the same $T, \varepsilon, d$)
10:        **if** $b_t^y \neq \tilde{y} \, \forall y \in \mathcal{Y}$ and $\exists a \in \mathcal{Y} : b_t^y = 1$ **then**
11:            Take any action $y$ with $b_t^y = 1$
12:        **else**
13:            Query the mentor

---

which occurs independently with probability $p$ on each time step. Thus standard Hedge run on $\hat{\ell}$ is the "counterpart without queries" of HEDGEWITHQUERIES.

**The rest of the proof.** The other elements of the proof of Lemma 3.5 are as follows:

1. Lemma 3.1, which analyzes the standard version of Hedge (i.e., no queries and losses are observed on every time step).

2. Applying Lemma 3.1 to a $\hat{\ell}$.

3. Arithmetic and rearranging terms.

The proof of Lemma 3.1 relies on simple arithmetic properties of the Hedge weights. Regardless of the adversary's behavior, $\hat{\ell}$ is a well-defined loss function, so Lemma 3.1 can be applied. Step 3 clearly has no dependence on the type of adversary. Thus we conclude that Lemma 3.5 extends to adaptive adversaries.

## C   GENERALIZING THEOREM 5.2 TO MANY ACTIONS

We use the standard "one versus rest" reduction (see, e.g., Chapter 29 of Shalev-Shwartz & Ben-David (2014)). For each action $y$, we will learn a binary classifier which predicts whether action $y$ is the mentor's action. Formally, for each $y \in \mathcal{Y}$, define the policy class $\Pi_y = \{\pi_y : \pi \in \Pi$ and $\pi_y(x) = \mathbf{1}(\pi(x) = y)) \, \forall x \in \mathcal{X}\}$. Informally, for each policy $\pi : \mathcal{X} \to \mathcal{Y}$ in $\Pi$, there exists a policy $\pi_y : \mathcal{X} \to \{0, 1\}$ in $\Pi_y$ such that $\pi_y(x) = \mathbf{1}(\pi(x) = y)$ for all $x \in \mathcal{X}$.

Algorithm 3 runs one copy of our binary-action algorithm Algorithm 1 for each action $y \in \mathcal{Y}$. At each time step $t$, the copy for action $y$ returns an action $b_t^y$, with $b_t^y = 1$ indicating a belief that $y = \pi^m(x_t)$ and $b_t^y = 0$ indicating a belief that $y \neq \pi^m(x_t)$. (Note that $b_t^y = \tilde{y}$ is also possible, indicating that the mentor was queried.)

The key idea is that if $b_t^y$ is correct for each action $y$, there will be exactly one $y$ such that $b_t^y = 1$, and specifically it will be $y = \pi^m(x_t)$. Thus we are guaranteed to take the mentor's action on such time steps. The analysis for Theorem 5.2 (specifically, Lemma B.1) bounds the number of time steps when a given copy of Algorithm 1 is incorrect, so by the union bound, the number of time steps where *any* copy is incorrect is $|\mathcal{Y}|$ times that bound. That in turn bounds the number of time steps where Algorithm 3 takes an action other than the mentor's. Similarly, the number of queries made by Algorithm 3 is at most $|\mathcal{Y}|$ times the bound from Theorem 5.2. The result is the following theorem:

**Theorem C.1.** *Assume $\pi^m \in \Pi$ where either (1) $\Pi_y$ has finite VC dimension $d$ and $\mathbf{x}$ is $\sigma$-smooth or (2) $\Pi_y$ has finite Littlestone dimension $d$ for all $y \in \mathcal{Y}$. Then for any $T \in \mathbb{N}$, Algorithm 3 with $T$ and $\varepsilon = T^{\frac{-2n}{2n+1}}$ satisfies*

$$\mathbb{E}[R_T] \in O\left(\frac{|\mathcal{Y}|dL}{\sigma} T^{\frac{-1}{2n+1}} \log T\right)$$

$$\mathbb{E}[|Q_T|] \in O\left(|\mathcal{Y}|T^{\frac{4n+1}{4n+2}}\left(\frac{d}{\sigma}\log T + \mathrm{diam}(\mathbf{x})^n\right)\right)$$

We use the following terminology and notation in the proof of Theorem C.1:

1. We refer to the copy of Algorithm 1 running on $\Pi_y$ as "copy $y$ of Algorithm 1".

2. Let $\pi_t^y$ and $X_t^y$ refer to the values of $\pi_t$ and $X_t$ for copy $y$ of Algorithm 1.

3. Let $\pi^{my} : \mathcal{X} \to \{0,1\}$ be the policy defined by $\pi^{my}(x) = \mathbf{1}(\pi^m(x_t) = y)$. Note that querying the mentor tells the agent $\pi^m(x_t)$, which allows the agent to compute $\pi^{my}(x_t)$: this is necessary when Algorithm 1 queries while running on some $\Pi_y$.

4. Let $V_T^y = \{t \in [T] : b_t^y \neq \pi^{my}(x_t)\}$ be the set of time steps where $\pi_t^y$ does not correctly determine whether the mentor would take action $y$ and let $V_T = \{t \in [T] : y_t \neq \pi^m(x_t)\}$ be the set of time steps where the agent's action doesn't match the mentor's.

**Lemma C.1.** *We have $|V_T| \leq \sum_{y \in \mathcal{Y}} |V_T^y|$.*

*Proof.* We claim that $V_T \subseteq \cup_{y \in \mathcal{Y}} V_T^y$. Suppose the opposite: then there exists $t \in V_T$ such that $b_t^y = \pi^{my}(x_t)$ for all $y \in \mathcal{Y}$. Since $\pi^m(x_t) \in \mathcal{Y}$, there is exactly one $y \in \mathcal{Y}$ such that $\mathbf{1}(\pi^m(x_t) = y) = \pi^{my}(x_t) = b_t^y = 1$. Specifically, this holds for $y = \pi^m(x_t)$. But then Algorithm 3 takes action $\min\{y \in \mathcal{Y} : b_t^y = 1\} = \pi^m(x_t)$, which contradicts $t \in V_T$. Therefore $V_T \subseteq \cup_{y \in \mathcal{Y}} V_T^y$, and applying the union bound completes the proof. $\square$

**Lemma C.2.** *For all $t \in [T]$, $\mu^m(x_t) - \mu(x_t, y_t) \leq L\varepsilon^{1/n}$.*

*Proof.* The argument is similar to the proof of Lemma B.2. If $\mu^m(x_t) \neq \mu(x_t, y_t)$, then $y_t = y$ for some $y \in \mathcal{Y}$ where $b_t^y = 1$. Therefore copy $y$ of Algorithm 1 did not query at time $t$ and $\pi_t^y(x_t) = 1$. Let $(x', y') = \arg\min_{(x,y) \in X_t^y : \pi_t^y(x_t) = y} ||x_t - x||$. Then $||x_t - x'|| \leq \varepsilon^{1/n}$ and $y' = \pi_t^y(x_t) = 1$.

By construction of $X_t^y$, $y' = \pi^{my}(x')$ so $\pi^{my}(x') = 1$ which implies $\pi^m(x') = y$. Then by the local generalization assumption,

$$\mu(x_t, y_t) = \mu(x_t, y) = \mu(x_t, \pi^m(x')) \geq \mu^m(x_t) - L||x_t - x'|| \geq \mu^m(x_t) - L\varepsilon^{1/n}$$

as required. $\square$

**Theorem C.1.** *Assume $\pi^m \in \Pi$ where either (1) $\Pi_y$ has finite VC dimension $d$ and $\mathbf{x}$ is $\sigma$-smooth or (2) $\Pi_y$ has finite Littlestone dimension $d$ for all $y \in \mathcal{Y}$. Then for any $T \in \mathbb{N}$, Algorithm 3 with $T$ and $\varepsilon = T^{\frac{-2n}{2n+1}}$ satisfies*

$$\mathbb{E}[R_T] \in O\left(\frac{|\mathcal{Y}|dL}{\sigma}T^{\frac{-1}{2n+1}}\log T\right)$$

$$\mathbb{E}[|Q_T|] \in O\left(|\mathcal{Y}|T^{\frac{4n+1}{4n+2}}\left(\frac{d}{\sigma}\log T + \mathrm{diam}(\mathbf{x})^n\right)\right)$$

*Proof.* Theorem 5.2 implies that each copy of Algorithm 1 makes $O\left(T^{\frac{4n+1}{4n+2}}\left(\frac{d}{\sigma}\log T + \mathrm{diam}(\mathbf{x})^n\right)\right)$ queries in expectation, so by linearity of expectation, the expected number of queries made by Algorithm 3 is $O\left(|\mathcal{Y}|T^{\frac{4n+1}{4n+2}}\left(\frac{d}{\sigma}\log T + \mathrm{diam}(\mathbf{x})^n\right)\right)$. Using the same argument as in the proof of Lemma B.7 (with Lemma C.2 replacing Lemma B.2), we get

$$R_T = \prod_{t=1}^T \mu^m(x_t) - \prod_{t=1}^T \mu(x_t, y_t)$$

$$\leq \sum_{t=1}^T \left(\mu^m(x_t) - \min(\mu^m(x_t), \mu(x_t, y_t))\right)$$

$$\leq |V_T|L\varepsilon^{1/n}$$

Then by Lemma C.1, $R_T \le L\varepsilon^{1/n} \sum_{y \in \mathcal{Y}} |V_T^y|$. Taking the expectation and applying Lemma B.1 to each $V_T^y$ gives us

$$\mathbb{E}[R_T] \le L\varepsilon^{1/n} \sum_{y \in \mathcal{Y}} O\left(\frac{d}{\sigma} T\varepsilon \log(1/\varepsilon) \log T\right) = O\left(|\mathcal{Y}| L\varepsilon^{1/n} \frac{d}{\sigma} T\varepsilon \log(1/\varepsilon) \log T\right)$$

as required. □

# D    THERE EXIST POLICY CLASSES WHICH ARE LEARNABLE IN OUR SETTING BUT NOT IN THE STANDARD ONLINE MODEL

This section presents another algorithm with subconstant regret and sublinear queries, but under different assumptions. The primary takeaway here is that our algorithm can handle the class of thresholds on $[0, 1]$, which is known to have infinite Littlestone dimension and thus be hard in the standard online learning model. (Example 21.4 in Shalev-Shwartz & Ben-David (2014)).

Specifically, we assume a 1D input space and we allow the input sequence to be fully adversarial chosen. Instead of VC/Littlestone dimension, we consider the following notion of simplicity:

**Definition D.1.** Given a mentor policy $\pi^m$, partition the input space $\mathcal{X}$ into intervals such that all inputs within each interval share the same mentor action. Let $\{X_1, \ldots, X_k\}$ be a partition that minimizes the number of intervals. We call each $X_j$ a *segment*. Let $f(\pi^m)$ denote the number of segments in $\pi^m$.

Bounding the number of segments is similar conceptually to VC dimension in that it limits the ability of the policy class to realize arbitrary combinations of labels (i.e., mentor actionx) on $\mathbf{x}$. For example, if $\Pi$ is the class of thresholds on $[0, 1]$, every $\pi \in \Pi$ has at most two segments, and thus the positive result in this section will apply. This demonstrates the existence of policy classes which are learnable in our setting but not learnable in the standard online learning model, meaning that the two settings do not exactly coincide.

We prove the following result:

**Theorem D.2.** *For any $\mathbf{x} \in \mathcal{X}^T$, any $\pi^m$ with $f(\pi^m) \le K$, and any function $g : \mathbb{N} \to \mathbb{N}$, Algorithm 4 makes at most $(\mathrm{diam}(\mathbf{x}) + 4)g(T)$ queries and satisfies $R_T \le \frac{2LKT}{g(T)^2}$.*

Choosing $g(T) = T^c$ for $c \in (1/2, 1)$ is sufficient to subconstant regret and sublinear queries:

**Theorem D.3.** *For any $c \in (1/2, 1)$, Algorithm 4 with $g(T) = T^c$ makes $O(T^c(\mathrm{diam}(\mathbf{x}) + 1))$ queries and satisfies*

$$\lim_{T \to \infty} \sup_{\mathbf{x} \in \mathcal{X}^T} \sup_{\mu} \sup_{\pi^m : f(\pi^m) \le K} R_T = 0$$

Our algorithm does not need to know $L$ or the number of segments; it only needs to know $T$.

## D.1    INTUITION BEHIND THE ALGORITHM

The algorithm maintains a set of buckets which partition the observed portion of the input space. Each bucket's length determines the maximum loss in payoff we will allow from that subset of the input space. As long as the bucket contains a query from a prior time step, local generalization allows us to bound $\mu^m(x_t) - \mu(x_t, y_t)$ based on the length of the bucket containing $x_t$. We always query if the bucket does not contain a prior query

The granularity of the buckets is controlled by a function $g$, with the initial buckets having length $1/g(T)$. Since we can expect one query per bucket, we need $g(T) \in o(T)$ to ensure sublinear queries.

Regardless of the bucket length, the adversary can still place multiple segments in the same bucket $B$. A single query only tells us the optimal action for one of those segments, so we risk a payoff as bad as $\mu^m(x_t) - O(\mathrm{len}(B))$ whenever we choose not to query. We can endure a limited number of such payoffs, but if we never query again in that bucket, we may suffer $\Theta(T)$ such payoffs. Letting $\mu^m(x_t) = 1$ for simplicity, that would lead to $\prod_{t=1}^T \mu(x_t, y_t) \le \left(1 - \frac{1}{O(g(T))}\right)^{\Theta(T)}$, which converges to 0 (i.e., guaranteed catastrophe) when $g(T) \in o(T)$.

**Algorithm 4** achieves subconstant regret when the mentor's policy has a bounded number of segments.

```
 1: function AVOIDCATASTROPHE(T ∈ ℕ, g : ℕ → ℕ)
 2:     X_Q ← ∅                                              ▷ Previously queried inputs
 3:     π ← ∅                                                ▷ Records π^m(x) for each x ∈ X_Q
 4:     ℬ ← ∅                                                ▷ The set of active buckets
 5:     for t from 1 to T do
 6:         EVALUATEINPUT(x_t)

 7: function EVALUATEINPUT(x ∈ 𝒳)
 8:     if s ∉ B for all B ∈ ℬ then    ▷ No bucket containing x: create a new bucket and try again
 9:         B ← [ (j-1)/g(T), j/g(T) ] for j ∈ ℤ such that x ∈ B
10:         ℬ ← ℬ ∪ {B}
11:         n_B ← 0                                          ▷ Number of time steps that have used B
12:         EVALUATEINPUT(x)
13:     else
14:         B ← any bucket containing x
15:         if X_Q ∩ B = ∅ then                             ▷ No queries in this bucket
16:             Query mentor and observe π^m(x)
17:             π(x) ← π^m(x)
18:             X_Q ← X_Q ∪ {x}
19:             n_B ← n_B + 1
20:         else if n_B < T/g(T) then       ▷ Bucket has a query and isn't full: take that action
21:             Let x' ∈ X_Q ∩ B
22:             Take action π(x')
23:             n_B ← n_B + 1
24:         else                                            ▷ Bucket is full: split bucket and try again
25:             B = [a, b]
26:             (B_1, B_2) ← ( [a, (a+b)/2], [(a+b)/2, b] )
27:             (x_{B_1}, x_{B_2}) ← (0, 0)
28:             ℬ ← ℬ ∪ {B_1, B_2} \ B
29:             EVALUATEINPUT(x)
```

This failure mode suggests a natural countermeasure: if we start to suffer significant (potential) losses in the same bucket, then we should probably query there again. One way to structure these supplementary queries is by splitting the bucket in half when enough time steps have involved that bucket. It turns out that splitting after $T/g(T)$ time steps is a sweet spot.

## D.2  NOTATION FOR THE PROOF

We will use the following notation throughout the proof of Theorem D.2:

- Let $V_T = \{t \in [T] : \mu(x_t, y_t) < \mu^m(x_t)\}$ be the set of time steps with a suboptimal payoff.
- Let $B_t$ be the bucket that is used on time step $t$ (as defined on line 14 of Algorithm 4).
- Let $d(B)$ be the *depth* of bucket $B$
  - Buckets created on line 9 are depth 0.
  - We refer to $B_1, B_2$ created on line 26 as the children of the bucket $B$ defined on line 14.
  - If $B'$ is the child of $B$, $d(B') = d(B) + 1$.
  - Note that $\text{len}(B) = \frac{1}{g(T)2^{d(B)}}$.
- Viewing the set of buckets are a binary tree defined by the "child" relation, we use the terms "ancestor" and "descendant" in accordance with their standard tree definitions.
- Let $\mathcal{B}_V = \{B : \exists t \in V_T \text{ s.t. } B_t = B\}$ be the set of buckets that ever produced a suboptimal payoff.
- Let $\mathcal{B}'_V = \{B \in \mathcal{B}_V : \text{no descendant of } B \text{ is in } \mathcal{B}_V\}$.

### D.3 PROOF ROADMAP

The proof proceeds in the following steps:

1. Bound the total number of buckets and therefore the total number of queries (Lemma D.1).

2. Bound the suboptimality on a single time step based on the bucket length and $L$ (Lemma D.2).

3. Bound the sum of bucket lengths on time steps where we make a mistake (Lemma D.4), with Lemma D.3 as an intermediate step. This captures the total amount of suboptimality.

4. As in the proof of Theorem 5.2, Lemma B.3 transforms the multiplicative objective into an additive form. Lemma D.5 bounds the additive objective using Lemmas D.2 and D.4.

5. Combining Lemmas D.5 and B.3 bounds the regret (Lemma D.6).

6. Theorem D.2 directly follows from Lemmas D.1 and D.6.

### D.4 PROOF

**Lemma D.1.** *Algorithm 4 performs at most $(\mathrm{diam}(\mathbf{x}) + 4)g(T)$ queries.*

*Proof.* Algorithm 4 performs at most one query per bucket, so the total number of queries is bounded by the total number of buckets. There are two ways to create a bucket: from scratch (line 9), or by splitting an existing bucket (line 26).

Since depth 0 buckets overlap only at their boundaries, and each depth 0 bucket has length $1/g(T)$, at most $g(T) \max_{t,t' \in [T]} |x_t - x_{t'}| = g(T) \mathrm{diam}(\mathbf{x})$ depth 0 buckets are subsets of the interval $[\min_{t \in [T]} x_t, \max_{t \in [T]} x_t]$. At most two depth 0 buckets are not subsets of that interval (one at each end), so the total number of depth 0 buckets is at most $g(T) \mathrm{diam}(\mathbf{x}) + 2$.

We split a bucket $B$ when $n_B$ reaches $T/g(T)$, which creates two new buckets. Since each time step increments $n_B$ for a single bucket $B$, and there are a total of $T$ time steps, the total number of buckets created via splitting is at most $\frac{2T}{T/g(T)} = 2g(T)$. Therefore the total number of buckets ever in existence is $(\mathrm{diam}(\mathbf{x}) + 2)g(T) + 2 \leq (\mathrm{diam}(\mathbf{x}) + 4)g(T)$, so Algorithm 4 performs at most $(\mathrm{diam}(\mathbf{x}) + 4)g(T)$ queries. $\square$

**Lemma D.2.** *For each $t \in [T]$, $\mu(x_t, y_t) \geq \mu^m(x_t) - L \mathrm{len}(B_t)$.*

*Proof.* If we query the mentor at time $t$, $\mu(x_t, y_t) = \mu^m(x_t)$. Thus assume we do not query the mentor at time $t$: then there exists $x' \in B_t$ (as defined on line 21 of Algorithm 4) such that $y_t = \pi(x') = \pi^m(x')$. Since $x_t$ and $x'$ are both in $B_t$, $|x_t - x'| \leq \mathrm{len}(B_t)$. Then by the local generalization assumption, $\mu(x_t, y_t) = \mu(x_t, \pi^m(x')) \geq \mu^m(x_t) - L||x_t - x'|| \geq \mu^m(x_t) - L \mathrm{len}(B_t)$. $\square$

**Lemma D.3.** *If $\pi^m$ has at most $K$ segments, $|\mathcal{B}'_V| \leq K$.*

*Proof.* Now consider any $B \in \mathcal{B}'_V$. By definition of $\mathcal{B}'_V$, there exists $t \in V_T$ such that $x_t \in B$. Then there exists $x' \in B$ (as defined in Algorithm 4) such that $y_t = \pi(x') = \pi^m(x')$. Since $t \in V_T$, we have $\pi^m(x_t) \neq y_t = \pi^m(x')$. Thus $x_t$ and $x'$ are in different segments, but are both in $B$. Therefore any $B \in \mathcal{B}'_V$ must intersect at least two segments. Since $B$ is an interval, if it intersects two segments, it must intersect two adjacent segments $X_j$ and $X_{j+1}$. Furthermore, $B$ must contain an open neighborhood centered on the boundary between $X_j$ and $X_{j+1}$.

Now consider some $B' \in \mathcal{B}'_V$ with $B \neq B'$. We $|B \cap B'| \leq 1$: otherwise one must be the descendant of the other, which contradicts the definition of $\mathcal{B}'_V$. Suppose $B'$ also intersects both $X_j$ and $X_{j+1}$: since $B'$ is also an interval, $B'$ must also contain an open neighborhood centered on the boundary between those two segments. But then $|B \cap B'| > 1$, which is a contradiction.

Therefore any pair of adjacent segments $X_j$ and $X_{j+1}$, there is at most one bucket in $\mathcal{B}'_V$ which contains an open neighborhood around their boundary. Since there are at most $K - 1$ pairs of adjacent segments, we have $|\mathcal{B}'_V| \leq K - 1 \leq K$. $\square$

**Lemma D.4.** *We have* $\sum_{t \in V_T} \operatorname{len}(B_t) \leq \frac{2KT}{g(T)^2}.$

*Proof.* For every $t \in V_T$, we have $B_t = B$ for some $B \in \mathcal{B}_V$, so

$$\sum_{t \in V_T} \operatorname{len}(B_t) = \sum_{B \in \mathcal{B}_V} \sum_{t \in V_T : B = B_t} \operatorname{len}(B_t)$$

Next, observe that every $B \in \mathcal{B}_V \setminus \mathcal{B}'_V$ must have a descendent in $\mathcal{B}'_V$: otherwise we would have $B \in \mathcal{B}'_V$. Let $\mathcal{A}(B)$ denote the set of ancestors of $B$, plus $B$ itself. Then we can write

$$\sum_{t \in V_T} \operatorname{len}(B_t) \leq \sum_{B' \in \mathcal{B}'_V} \sum_{B \in \mathcal{A}(B')} \sum_{t \in V_T : B = B_t} \operatorname{len}(B_t)$$

$$= \sum_{B' \in \mathcal{B}'_V} \sum_{B \in \mathcal{A}(B')} |\{t \in V_T : B = B_t\}| \cdot \operatorname{len}(B_t)$$

For any bucket $B$, the number of time steps $t$ with $B = B_t$ is at most $T/g(T)$. Also recall that $\operatorname{len}(B) = \frac{1}{g(T)2^{d(B)}}$. Therefore

$$\sum_{B \in \mathcal{A}(B')} \frac{|\{t \in V_T : B = B_t\}|}{g(T)2^{d(B)}} \leq \frac{T}{g(T)^2} \sum_{B \in \mathcal{A}(B')} \frac{1}{2^{d(B)}}$$

$$= \frac{T}{g(T)^2} \sum_{d=0}^{d(B')} \frac{1}{2^d} \leq \frac{T}{g(T)^2} \sum_{d=0}^{\infty} \frac{1}{2^d} = \frac{2T}{g(T)^2}$$

Then by Lemma D.3,

$$\sum_{t \in V_T} \operatorname{len}(B_t) \leq \sum_{B' \in \mathcal{B}'_V} \frac{2T}{g(T)^2} = \frac{2T|\mathcal{B}'_V|}{g(T)^2} \leq \frac{2KT}{g(T)^2}$$

as claimed. $\qquad\square$

**Lemma D.5.** *Under the conditions of Theorem D.2, Algorithm 4 satisfies*

$$\sum_{t=1}^{T} \left( \mu^m(x_t) - \min(\mu^m(x_t), \mu(x_t, y_t)) \right) \leq \frac{2LKT}{g(T)^2}$$

*Proof.* For $t \notin V_T$ we have $\min(\mu^m(x_t), \mu(x_t, y_t)) = \mu^m(x_t)$ by definition, and Lemma D.2 implies that $\min(\mu^m(x_t), \mu(x_t, y_t)) \geq L \operatorname{len}(B_t)$ for all $t \in [T]$. Thus

$$\sum_{t=1}^{T} \left( \mu^m(x_t) - \min(\mu^m(x_t), \mu(x_t, y_t)) \right) \leq \sum_{t \in V_T} \left( \mu^m(x_t) - \min(\mu^m(x_t), \mu(x_t, y_t)) \right) \leq L \sum_{t \in V_T} \operatorname{len}(B_t)$$

Then by Lemma D.4,

$$\sum_{t=1}^{T} \left( \mu^m(x_t) - \min(\mu^m(x_t), \mu(x_t, y_t)) \right) \leq \frac{2LKT}{g(T)^2}$$

as required. $\qquad\square$

**Lemma D.6.** *Under the conditions of Theorem D.2, Algorithm 4 satisfies* $R_T \leq \frac{2LKT}{g(T)^2}$.

*Proof.* Let $a_t = \mu^m(x_t)$ and $b_t = \min(\mu^m(x_t), \mu(x_t, y_t))$ for all $t \in [T]$. Then by Lemma B.3,

$$\prod_{t=1}^{T} \mu^m(x_t) - \prod_{t=1}^{T} \min(\mu^m(x_t), \mu(x_t, y_t)) \leq \sum_{t=1}^{T} \left( \mu^m(x_t) - \min(\mu^m(x_t), \mu(x_t, y_t)) \right)$$

Since $\mu(x_t, y_t) \geq \min(\mu^m(x_t), \mu(x_t, y_t))$ for all $t \in [T]$, we have

$$R_T = \prod_{t=1}^{T} \mu^m(x_t) - \prod_{t=1}^{T} \mu(x_t, y_t) \leq \sum_{t=1}^{T} \left( \mu^m(x_t) - \min(\mu^m(x_t), \mu(x_t, y_t)) \right)$$

Applying Lemma D.5 completes the proof. $\qquad\square$

Theorem D.2 follows from Lemma D.1 and Lemma D.6.

## E  OTHER PROOFS

Proposition E.1 states that Lipschitz continuity implies local generalization when the mentor is optimal.

**Proposition E.1.** *Assume that for all $x, x' \in \mathcal{X}$ and $y \in \mathcal{Y}$, $|\mu(x, a) - \mu(x', a)| \leq L||x - x'||$. Also assume that $\mu(x, \pi^m(x)) = \max_{y \in \mathcal{Y}} \mu(x, y)$ for all $x \in \mathcal{X}$. Then $\pi^m$ satisfies local generalization with constant $2L$.*

*Proof.* For any $x, x' \in \mathcal{X}$, we have

$$
\begin{aligned}
\mu(x, \pi^m(x')) &\geq \mu(x', \pi^m(x')) - L||x - x'|| && \text{(Lipschitz continuity of } \mu) \\
&\geq \mu(x', \pi^m(x)) - L||x - x'|| && (\pi^m \text{ is optimal for } x') \\
&\geq \mu(x, \pi^m(x)) - 2L||x - x'|| && \text{(Lipschitz continuity of } \mu \text{ again)} \\
&= \mu^m(x) - 2L||x - x'|| && \text{(Definition of } \mu^m(x))
\end{aligned}
$$

Since $\pi^m$ is optimal for $x$, we have

$$
\mu^m(x) + 2L||x - x'|| \geq \mu^m(x) \geq \mu(x, \pi^m(x'))
$$

Thus $-2L||x - x'|| \leq \mu(x, \pi^m(x')) - \mu^m(x) \leq 2L||x - x'||$. This is equivalent to $|\mu(x, \pi^m(x')) - \mu^m(x)| \leq 2L||x - x'||$, completing the proof.

$\square$

Proposition E.2 states that the achievability of subconstant regret does not depend on whether we require expected sublinear queries or worst-case sublinear queries.

**Proposition E.2.** *Suppose an algorithm satisfies $\lim_{T \to \infty} \sup_{\mu, \pi^m} \mathbb{E}[R_T] = 0$ and $\sup_{\mu, \pi^m} \mathbb{E}[|Q_T|] \in o(T)$. Then there exists $h : \mathbb{N} \to \mathbb{N}$ such that (1) $h(T) \in o(T)$ and (2) if the algorithm is modified to simply stop querying if the number of queries reaches $h(T)$, the algorithm still satisfies $\lim_{T \to \infty} \sup_{\mu, \pi^m} \mathbb{E}[R_T] = 0$.*

*Proof.* We use $Q_T, R_T$ to refer to the queries and regret of the original algorithm, and $Q'_T, R'_T$ to refer to the queries and regret of the modified algorithm.

Since $\sup_{\mu, \pi^m} \mathbb{E}[|Q_T|] \in o(T)$, there exists $g : \mathbb{N} \to \mathbb{N}$ such that $\sup_{\mu, \pi^m} \mathbb{E}[|Q_T|] \leq g(T)$ and $g(T) \in o(T)$. Let $h(T) = \sqrt{g(T)T}$; then $h(T) \in o(T)$ by Lemma A.1. Markov's inequality implies that

$$
\Pr[|Q_T| > h(T)] \leq \frac{\mathbb{E}[|Q_T|]}{h(T)} \leq \frac{g(T)}{\sqrt{g(T)T}} = \sqrt{\frac{g(T)}{T}}
$$

Let $\xi$ denote the event that at some point, the original algorithm would query, but the modified algorithm cannot because $|Q'_T| = h(T)$. Then $\Pr[\xi] \leq \Pr[|Q_T| > h(T)]$ (the inequality is because the modified algorithm might not want to query more anyway). Also note that the algorithms are equivalent if $\xi$ does not occur, so $\mathbb{E}[R'_T \mid \neg\xi] = \mathbb{E}[R_T]$. Hence

$$
\begin{aligned}
\mathbb{E}[R'_T] &= \mathbb{E}[R'_T \mid \neg\xi]\Pr[\neg\xi] + \mathbb{E}[R'_T \mid \xi]\Pr[\xi] \\
&\leq \mathbb{E}[R_T] \cdot 1 + 1 \cdot \Pr[\xi] \\
&\leq \mathbb{E}[R_T] + \sqrt{\frac{g(T)}{T}}
\end{aligned}
$$

Since $g(T) \in o(T)$, we get

$$
\lim_{T \to \infty} \sup_{\mu, \pi^m} \mathbb{E}[R'_T] \leq \lim_{T \to \infty} \sup_{\mu, \pi^m} \left( \mathbb{E}[R_T] + \sqrt{\frac{g(T)}{T}} \right)
$$

$$
= \lim_{T \to \infty} \sup_{\mu, \pi^m} \mathbb{E}[R_T] + \lim_{T \to \infty} \sqrt{\frac{g(T)}{T}}
$$

$$= 0$$

as required. □

Theorem E.3 shows that avoiding catastrophe is impossible without local generalization, even when $\mathbf{x}$ is $\sigma$-smooth and $\Pi$ has finite VC dimension. The first insight is that without local generalization, we can define $\mu(x, y) = \mathbf{1}(y = \pi^m(x))$ so that a single mistake causes $\prod_{t=1}^{T} \mu(x_t, y_t) = 0$. To lower bound $\Pr\left[\prod_{t=1}^{T} \mu(x_t, y_t) = 0\right]$, we use a similar approach to the proof of Theorem 4.1: divide $\mathcal{X} = [0, 1]$ into $f(T)$ independent sections with $|Q_T| << f(T) << T$, so that the agent can only query a small fraction of these sections. However, the proof of Theorem E.3 is a bit easier, since we only need the agent to make a single mistake.

The proof of Theorem E.3 assumes sublinear queries unconditionally, but recall from Proposition E.2 that the distinction between worst-case sublinear queries and expected sublinear queries is not significant.

**Theorem E.3.** *Let $\mathcal{X} = [0, 1]$ and $\mathcal{Y} = \{0, 1\}$. Assume each input is sampled i.i.d. from the uniform distribution on $\mathcal{X}$ and define the mentor policy class by the set of intervals within $\mathcal{X}$, i.e., $\Pi = \{\pi : \exists a, b \in [0, 1] \ s.t \ \pi(x) = \mathbf{1}(x \in [a, b]) \ \forall x \in \mathcal{X}\}$. Then without the local generalization assumption, any algorithm with sublinear queries satisfies $\lim_{T \to \infty} \sup_{\mu, \pi^m} \mathbb{E}[R_T] = 1$.*

*Proof.* **Part 1: Setup.** Consider any algorithm which makes sublinear worst-case queries: then there exists $g : \mathbb{N} \to \mathbb{N}$ where $\sup_{\mu, \pi^m} |Q_T| \le g(T)$ and $g(T) \in o(T)$. Define $f(T) := \sqrt{(g(T) + 1)T}$; by Lemma A.1, $g(T) \in o(f(T))$ and $f(T) \in o(T)$. Divide $\mathcal{X}$ into $f(T)$ equally sized sections $X_1, \ldots, X_{f(T)}$ in the exactly the same way as in Section 4.2; see also Figure 2. Assume that each $x_t$ is in exactly one section: this assumption holds with probability 1, so it does not affect the regret.

We use the probabilistic method: sample a segment $j^m \in [f(T)]$ uniformly at random, define $\pi^m$ by $\pi^m(x) = \mathbf{1}(x \in X_{j^m})$, and define $\mu$ by $\mu(x, y) = \mathbf{1}(y = \pi^m(x))$. In words, the mentor takes action 1 iff the input is in section $j^m$, and the agent receives payoff 1 if its action matches the mentor's and zero otherwise. Since any choice of $j^m$ defines a valid $\mu$ and $\pi^m$, we have

$$\sup_{\mu, \pi^m} \mathbb{E}_{\mathbf{x}, \mathbf{y}} \left[R_T(\mathbf{x}, \mathbf{y}, \mu, \pi^m)\right] \ge \mathbb{E}_{j^m} \mathbb{E}_{\mathbf{x}, \mathbf{y}} \left[R_T(\mathbf{x}, \mathbf{y}, \mu, \pi^m)\right]$$

Let $J_{\neg Q} = \{j \in [f(T)] : x_t \notin X_j \ \forall t \in Q_T\}$ be the set of sections which are never queried. Let $j_1, \ldots, j_k$ be the sequence of sections queried by the agent: then $k \le |Q_T| \le g(T)$.

**Part 2: The agent is unlikely to determine $j^m$.** By the chain rule of probability,

$$\Pr[j^m \in J_{\neg Q}] = \Pr\left[j_i \ne j^m \ \forall i\right] = \prod_{i=1}^{k} \Pr\left[j_i \ne j^m \mid j_r \ne j^m \ \forall r < i\right]$$

Now fix $i$ and assume $j_r \ne j^m \ \forall r < i$. Queries in sections other than $j^m$ provide no information about the value of $j^m$, so $j^m$ is uniformly distributed across the set of sections not yet queried, i.e., $\{j \in [f(T)] : j_r \ne j \ \forall r < i\}$. There are at least $f(T) - i + 1$ such sections, since there are $i - 1$ prior queries at this point. Thus $\Pr[j_i \ne j^m \mid j_r \ne j^m \ \forall r < i] \le \frac{f(T) - i}{f(T) - i + 1}$ (the inequality is because it could also be 0 if $j_i = j_r$ for some $i < r$). Therefore

$$\begin{aligned}
\Pr\left[j^m \in J_{\neg Q}\right] &\le \prod_{i=1}^{k} \frac{f(T) - i}{f(T) - i + 1} \\
&= \frac{f(T) - 1}{f(T)} \cdot \frac{f(T) - 2}{f(T) - 1} \cdots \frac{f(T) - k + 1}{f(T) - k + 2} \cdot \frac{f(T) - k}{f(T) - k + 1} \\
&= \frac{f(T) - k}{f(T)} \\
&\ge 1 - \frac{g(T)}{f(T)}
\end{aligned}$$

**Part 3: If the agent fails to determine $j^m$, it is likely to make at least one mistake.** For each $j \in J_{\neg Q}$, let $V_j = \{t \in [T] : x_t \in X_j\}$ be the set of time steps with inputs in section $j$. By

Lemma A.3, $\Pr[|V_{j^m}| = 0] \leq \exp\left(\frac{T}{16f(T)}\right)$. Then by the union bound, $\Pr[j^m \in J_{\neg Q} \text{ and } |V_{j^m}| > 0] \geq 1 - \frac{g(T)}{f(T)} - \exp\left(\frac{-T}{16f(T)}\right)$. For the rest of Part 3, assume $j^m \in J_{\neg Q}$ and $|V_{j^m}| > 0$.

Since $j^m \in J_{\neg Q}$, the agent has no information about $j^m$ other than that it is in $J_{\neg Q}$. This means that for all $j \in J_{\neg Q}$ and $t \in V_j$, $j^m$ is conditionally (under the condition of $j^m \in J_{\neg Q}$) independent of $y_t$. We proceed by case analysis.

Case 1: For all $j \in J_{\neg Q}, t \in V_j$, we have $y_t = 0$. In particular, this holds for $j = j^m$, and we know there exists at least one $t \in V_{j^m}$ since $|V_{j^m}| > 0$. Then $y_t \neq \pi^m(x_t)$, so $\mu(x_t, y_t) = 0$ and thus $\Pr\left[\prod_{r=1}^T \mu(x_r, y_r) = 0 \mid j^m \in J_{\neg Q} \text{ and } |V_{j^m}| > 0\right] = 1$.

Case 2: There exists $j \in J_{\neg Q}, t \in V_j$ with $y_t = 1$. Then $\mu(x_t, y_t) = 0$ unless $j = j^m$, so

$$\Pr\left[\prod_{r=1}^T \mu(x_r, y_r) = 0 \mid j^m \in J_{\neg Q} \text{ and } |V_{j^m}| > 0\right] \geq \Pr\left[\mu(x_t, y_t) = 0 \mid j^m \in J_{\neg Q} \text{ and } |V_{j^m}| > 0\right]$$

$$= \Pr\left[j \neq j^m \mid j^m \in J_{\neg Q} \text{ and } |V_{j^m}| > 0\right]$$

Conditioned on $j^m \in J_{\neg Q}$, $j^m$ is uniformly distributed across $J_{\neg Q}$, so

$$\Pr\left[\prod_{r=1}^T \mu(x_r, y_r) = 0 \mid j^m \in J_{\neg Q} \text{ and } |V_{j^m}| > 0\right] \geq 1 - \frac{1}{|J_{\neg Q}|} \geq 1 - \frac{1}{f(T) - g(T)}$$

Combining Case 1 and Case 2, we get the overall bound of

$$\Pr\left[\prod_{t=1}^T \mu(x_t, y_t) = 0 \mid j^m \in J_{\neg Q} \text{ and } |V_{j^m}| > 0\right] \geq 1 - \frac{1}{f(T) - g(T)}$$

and thus

$$\Pr\left[\prod_{t=1}^T \mu(x_t, y_t) = 0\right] \geq \Pr\left[\prod_{t=1}^T \mu(x_t, y_t) = 0 \text{ and } j^m \in J_{\neg Q} \text{ and } |V_{j^m}| > 0\right]$$

$$= \Pr\left[\prod_{t=1}^T \mu(x_t, y_t) = 0 \mid j^m \in J_{\neg Q} \text{ and } |V_{j^m}| > 0\right] \cdot \Pr\left[j^m \in J_{\neg Q} \text{ and } |V_{j^m}| > 0\right]$$

$$\geq \left(1 - \frac{1}{f(T) - g(T)}\right)\left(1 - \frac{g(T)}{f(T)} - \exp\left(\frac{-T}{16f(T)}\right)\right)$$

For brevity, let $\alpha(T)$ denote this final bound. Since $g(T) \in o(f(T))$ and $f(T) \in o(T)$, we have

$$\lim_{T \to \infty} \alpha(T) = \lim_{T \to \infty} \left(1 - \frac{1}{f(T) - g(T)}\right)\left(1 - \frac{g(T)}{f(T)} - \exp\left(\frac{-T}{16f(T)}\right)\right)$$

$$= (1 - 0)(1 - 0 - 0)$$

$$= 1$$

**Part 4: Putting it all together.** Since $\prod_{t=1}^T \mu(x_t, y_t) \leq 1$ always, we have

$$\mathbb{E}_{j^m} \mathbb{E}_{\mathbf{x},\mathbf{y}} \left[\prod_{t=1}^T \mu(x_t, y_t)\right] = \mathbb{E}_{j^m} \mathbb{E}_{\mathbf{x},\mathbf{y}} \left[\prod_{t=1}^T \mu(x_t, y_t) \mid \prod_{t=1}^T \mu(x_t, y_t) = 0\right] \cdot \Pr\left[\prod_{t=1}^T \mu(x_t, y_t) = 0\right]$$

$$+ \mathbb{E}_{j^m} \mathbb{E}_{\mathbf{x},\mathbf{y}} \left[\prod_{t=1}^T \mu(x_t, y_t) \mid \prod_{t=1}^T \mu(x_t, y_t) \neq 0\right] \cdot \Pr\left[\prod_{t=1}^T \mu(x_t, y_t) \neq 0\right]$$

$$\leq 0 \cdot \Pr\left[\prod_{t=1}^T \mu(x_t, y_t) = 0\right] + 1 \cdot \left(1 - \Pr\left[\prod_{t=1}^T \mu(x_t, y_t) = 0\right]\right)$$

$$\leq 1 - \alpha(T)$$

Since $\prod_{t=1}^{T} \mu^m(x_t) = 1$ always, we have

$$\sup_{\mu, \pi^m} \mathbb{E}_{\mathbf{x}, \mathbf{y}} \left[ R_T(\mathbf{x}, \mathbf{y}, \mu, \pi^m) \right] \geq \mathbb{E}_{j^m} \mathbb{E}_{\mathbf{x}, \mathbf{y}} \left[ R_T(\mathbf{x}, \mathbf{y}, \mu, \pi^m) \right]$$

$$= 1 - \mathbb{E}_{j^m} \mathbb{E}_{\mathbf{x}, \mathbf{y}} \left[ \prod_{t=1}^{T} \mu(x_t, y_t) \right]$$

$$\geq \alpha(T)$$

Therefore $\lim_{T \to \infty} \sup_{\mu, \pi^m} \mathbb{E}[R_T] \geq \lim_{T \to \infty} \alpha(T) = 1$, as required. $\quad\square$

