# OpenReview forum: "Avoiding Catastrophe in Online Learning by Asking for Help"
_ICLR.cc/2025/Conference — Submitted to ICLR 2025_

### Official Review · Reviewer_E2Pc · 2024-10-27

**Soundness:** 3
**Presentation:** 3
**Contribution:** 3
**Rating:** 6
**Confidence:** 2

**Summary:**

This paper proposes a novel online learning framework in which the learner aims to minimize the catastrophe probability, that is, avoid choosing catastrophic actions. In order to do that, it is assumed the possibility to query a mentor (the baseline) at time $t\in[T]$, which returns the mentor's action associated with the current state $s_t$. Moreover, local generalization of the mentor policy is assumed (namely, a sort of continuity assumption between states given the mentor policy). The performance metrics used to evaluate the online algorithms are the regret between the product of the avoiding catastrophe probabilities attained by the learner and the mentor and the number of queries performed. First, the authors show the impossibility of learning in the general setting. Thus, the authors focus on the case of finite VC or Littlestone dimension. In such a setting, the authors provide a simple hedge-kind of algorithm that attains sub constant regret and a sublinear number of queries to the mentor during the learning dynamic.

**Strengths:**

I believe that the setting proposed in this work is of interest from a theoretical perspective.
Indeed, to the best of my knowledge, the framework is novel, and as pointed out in the work, it presents some peculiar theoretical challenges. The results seem correct and reasonable to me. Finally, the authors put much effort into explaining the main ideas behind the theoretical results.

**Weaknesses:**

To me, there are two main weaknesses in this work. The first one is the lack of particular algorithmic novelty. The algorithm proposed is a clear, simple adaptation of hedge. Moreover, this work strongly relies on many different results from statistical learning theory and online learning (see Section 5.2, where all the results belong to existing works). This is indeed not a sufficient reason for rejection, but somehow, it worsens the contribution of the work.

The second weakness concerns the practical relevance of the setting. While I agree that the setting is interesting from a theoretical perspective, it is not clear to me why it should be of interest in real-world scenarios. Specifically, the authors clearly state that this setting is of practical relevance since, unlike standard online learning, it allows for the avoidance of catastrophic actions. In contrast, the standard exploration-exploitation trade-off requires, in general, to try all the possible actions, even really dangerous ones. Nevertheless, there exist many works on learning in the presence of **unknown** constraints which exactly tackle the problem specified above. While without further assumptions, it is not possible to satisfy unknown constraints in every round, it is sufficient to require a feasible solution in input, to be simultaneously no-regret w.r.t. the rewards and avoiding possible dangerous actions with high probability (see, e.g., [Liu et al. 2021], [Stradi et al. 2024] for online learning in MDPs). Moreover, notice that those papers do not need any query to the mentor, and as previously specified, do not focus only on avoiding bad actions, but they maximize specific unknown rewards functions. Finally, while their regret is sublinear in $T$, their regret definition seems to be much stronger since the baseline is the optimal solution and not a general mentor policy.


[Liu et al., 2021] "Learning policies with zero or bounded constraint violation for constrained mdps."

[Stradi et al. 2024] "Learning Adversarial MDPs with Stochastic Hard Constraints"

**Questions:**

Could you please discuss on the second weakness? Which are the main advantages of your setting w.r.t. the one described above?

---

> ### Author Response · Authors · 2024-11-14
> **Response to Reviewer E2Pc**
>
> We thank the reviewer for helpful comments. We respond to each of the weaknesses mentioned.
>
> 1. **Lack of algorithmic novelty**. Although we agree that the algorithm is quite simple, this can also be seen as an advantage, since simple algorithms are both more accessible to readers and more likely to be used in practice. Also, although our analysis utilizes many prior results, we believe that the packing technique we use to bound the number of queries is novel. Essentially, we have provided a technique for bounding the number of data points needed to cover a particular set with respect to the realized actions of the algorithm. For merely covering the set without regard for the algorithm’s actions, existing packing/covering number bounds would suffice, but our technique may be useful in contexts where a more refined packing argument is needed than one which deals with all data points in aggregate. See also lines 508-514 in the paper.
>
> 2. **Related work on CMDPs**.  The two linked papers study a related but distinct problem. Both papers assume that the learner knows a safe policy (what the reviewer calls “a feasible solution”) upfront, and ask whether the learner can obtain high reward while also remaining safe. In our setting, the learner has no prior knowledge about which actions are safe, and must learn a safe policy from scratch. In this sense, our work complements theirs: our goal is essentially to learn the baseline safe policy that their algorithms require. One can also view our problem as the “pessimistic” model and their problem as the “optimistic” model, with some applications better captured by our model while other applications are better captured by theirs. We will add a discussion of this related work to our paper.

---

> > ### Comment · Reviewer_E2Pc · 2024-11-15
> >
> > I would like to thank the Authors for their response. I will keep my current score.

---

> > > ### Author Response · Authors · 2024-11-18
> > > **Revised PDF now available**
> > >
> > > We have now uploaded a revised version of the paper which incorporates the feedback from all reviewers. We believe that this revision fully addresses the second weakness and partly addresses the first weakness raised by Reviewer E2Pc. Changes are highlighted in red. The relevant changes for this reviewer's feedback can be found on the following line numbers in the revised PDF:
> > >
> > > W1: Expanded the description of the novel packing argument (lines 511-515). However, the evaluation of technical novelty remains subjective.
> > >
> > > W2: Added a comparison to the CMDP literature and to Liu et al. [2021] and Stradi et al. [2024] in particular (lines 173-181).
> > >
> > > We understand that you have already decided to maintain your score, however we felt that we should update you along with the other reviewers for completeness.

---

### Official Review · Reviewer_kZKp · 2024-11-02

**Soundness:** 3
**Presentation:** 3
**Contribution:** 2
**Rating:** 5
**Confidence:** 3

**Summary:**

This paper considers a new variant of the classical online learning setting in which the aim is to avoid "catastrophe" relative to a mentor policy \pi^m. Concretely, letting \mu denote the reward, the aim is to choose a sequence of actions a_1, ..., a_T based on a sequence of covariates s_1, ..., s_T such that the following notion of regret is small:

\prod_{t=1}^{T}\mu(s_t, pi^m(s_t)) - \prod_{t=1}^{T}\mu(s_t, a_t).

That is, compared to the standard online learning formulation (e.g., Cesa-Bianchi and Lugosi), which considers regret of the form \sum_{t=1}^{T}\mu(s_t, pi^m(s_t)) - \sum_{t=1}^{T}\mu(s_t, a_t), we care about the product of rewards instead of the sum of rewards. This reflects the fact that we are averse to catastrophe, since a very small reward at any round $t$ can completely ruin our prospects of achieving low regret.

Beyond the differences above, the setting the authors consider also has the feature that the reward is never observed. Rather, the learner can choose to query the mentor at each round $t$ and observe pi^m(s_t). The aim is to ensure the regret and the number of mentor queries is sublinear in $T$.

The authors provide the following results:
* When (1) the mentor satisfies a "local generalization" property related to Lipschitzness and (2) their policy belongs to a class \Pi which is either a Littlestone class or a VC class (with the additional assumption that outcomes are "smooth") it is possible to achieve sublinear regret and expert queries through a variant of the exponential weights algorithm.

* Without the Littlestone or VC assumptions, sublinear regret is impossible, even under the local generalization assumption.

**Strengths:**

This paper makes a reasonable contribution to the online learning literature by proposing what is, to my knowledge, a novel setting and problem formulation. I believe the upper and lower bounds the authors provide are novel and do not necessarily follow from existing work. I also found the paper to be generally well written and easy to follow.

**Weaknesses:**

The main drawbacks preventing me from giving a higher score are as follows:

* While the problem formulation is loosely inspired by literature on AI risk, it is unclear whether the new problem formulation the authors provide actually has implications for this literature. Spelling out potential connections and implications of the results seems important, as otherwise I am worried the impact might be rather limited (e.g., a niche setting of interest to online learning experts, but not necessarily of interesting outside the core theory community). Overall, if there are interesting consequences of algorithic results here it would be great to highlight this.

* Given that the main contribution of the paper is to introduce a new problem formulation, it seems important to justify all of the assumptions and argue that they are fundamental. Here, I am somewhat concerned about the local generalization assumption, which seems a bit arbitrary and inelegant. While I can believe that some assumption of this type might be required, it is not clear that this specific one is fundamental. In particular, I do not like that the assumption requires that $s$ lies in Euclidean space, which feels quite arbitrary and not really in the spirit of online learning (which is generally agnostic to the space of covariates, and depends on it only through assumptions on the class \Pi itself). Nailing down what is the right notion of local generalization would definitely strengthen the paper.

-----

Some minor comments:
* The authors may want to compare with the literature on online learning with the logarithmic loss. E.g., if we do standard online learning with additive regret, but choose \log(\mu(s_t, a)) as the reward, then a standard additive regret bound of the form \sum_t reward(s_t, \pi^m) - reward(s_t, a_t) \leq Reg implies that \log(\prod \mu(s_t, pi^m)/\mu(s_t, a_t)) \leq Reg. This is a different guarantee from what the authors provide, but has a similar flavor, so it might be interesting to compare.

* This setting also has an imitation learning flavor, so it would be good to add some discussion on how it connects to the IL literature.

* The term "state" is a bit confusing/misleading since this is a standard online learning setting as opposed to RL (i.e., we just take the states as given, and are not interesting in counterfactuals wrt how they might have evolved if we had acted differently). Calling them "contexts" or "covariates" would be more clear.

**Questions:**

Is the local generalization assumption fundamental? See comments above

---

> ### Author Response · Authors · 2024-11-14
> **Response to Reviewer kZKp**
>
> We thank the reviewer for helpful comments. We respond to each of the weaknesses mentioned and also to the reviewer’s question (which corresponds to one of the weaknesses).
>
> 1. **Connections to AI risk**. The AI risk literature discusses in depth the importance of safety guarantees, but has had comparatively little success in formally establishing those guarantees. In particular, to our knowledge, we are the first to formally guarantee avoidance of catastrophe with a computationally tractable algorithm (e.g., without Bayesian inference). Although we do not claim that our model perfectly captures all real-world safety concerns, our model is quite flexible, allowing a wide range of state/context sequences, mentor policies, payoff functions, etc. We view our work not just as a stylized model of catastrophe, but as a real step towards the practical safety guarantees sought by the AI risk literature. If future work extends our results to not only avoid catastrophe but also guarantee high reward (e.g., in an MDP), one can start thinking about using such algorithms in practice.
>
> 2.  **Local generalization**. We agree with the reviewer on the importance of justifying assumptions, and we appreciate the opportunity to respond. Local generalization states that if the mentor told us that an action is safe in state $s’$, the action is likely also safe in a similar state $s$. To be precise, $\mu(s, \pi^m(s))$ is the payoff from taking the actual safe action in the current state $s$, while $\mu(s, \pi^m(s’))$ is the payoff from using what we learned in a previous state $s’$ in $s$. Local generalization states that the gap between these is bounded proportional to the similarity between those states: $|\mu(s,\pi^m(s)) - \mu(s, \pi^m(s’))| \le L||s-s’||$. The concept of borrowing knowledge from similar experiences seems fundamental to learning and is well-understood in the psychology literature ([example](https://link.springer.com/article/10.1007/s00426-023-01800-4)) and education literature ([example](https://files.eric.ed.gov/fulltext/EJ1217940.pdf)).
>
>     Crucially, our state space is not necessarily the 3D orientation of the agent, but can be any feature embedding of the agent’s situation. This allows for more abstract notions of "similarity". Also, our algorithm does not require knowledge of the feature embedding (see lines 406-409) and does not need to know $L$, so it suffices that there exists _some_ feature embedding which satisfies local generalization. The agent does not even need to know which embedding it is.
>
>     Finally, note that in the case of an optimal mentor, local generalization implies the more familiar Lipschitz continuity (Proposition E.1). The assumption of Lipschitz continuity is common in the bandit literature (see Chapter 4 of this [book](https://arxiv.org/pdf/1904.07272)).
>
> 3. **Euclidean space**. The reviewer makes a fair point regarding our Euclidean space assumption and the spirit of online learning. However, this assumption is actually not necessary: all of our results go through for a general metric space, with the difference that the regret would then depend explicitly on the packing number. We thought this would be less intuitive for readers since the subconstant nature of our regret bound is less obvious with the packing number present, and we also think that most real-world applications would involve feature embeddings in $\mathbb{R}^n$. However, we can switch our analysis to a general metric space if the reviewer thinks the trade-off is worth it.
>
> 4. **Online learning with log loss**. We agree that this literature is relevant, and thank the reviewer for pointing it out. The key difference is that those regret bounds are **sublinear** in T while our regret bound is **subconstant** in T. In other words, the help of a mentor and local generalization can reduce the regret by an entire factor of T. We will add a discussion of this to the paper.
>
> 5. **Imitation learning**. Thank you for the pointer; we will also add a discussion of imitation learning.
>
> 6. **The term “state”**. We agree with this point and ask whether the reviewer would be okay with the term “input”, which we hope is intuitive to readers from both online learning and RL.

---

> > ### Author Response · Authors · 2024-11-18
> > **Revised PDF now available**
> >
> > We have now uploaded a revised version of the paper which incorporates the feedback from all reviewers. We believe that this revision addresses each of the points raised by Reviewer kZKp, except for Concerns 1 and 6 (using the numbering in our response message). We will also revise the paper to address Concerns 1 and 6, but we were hoping to receive further feedback from the reviewer first.
> >
> > Changes are highlighted in red. The relevant changes for this reviewer's feedback can be found on the following line numbers in the revised PDF:
> >
> > Concern 2: Improved the discussion of justification of local generalization (lines 240-251)
> >
> > Concern 3: Mentioned that our assumption of Euclidean space is only for convenience (line 215)
> >
> > Concern 4: Added a discussion of online learning with log loss (lines 192-196)
> >
> > Concern 5: Mentioned imitation learning under related work (lines 206-207)
> >
> > Please let us know if this adequately addresses Concerns 2-5. If not, we would welcome further discussion.

---

> > ### Comment · Reviewer_kZKp · 2024-11-23
> >
> > Thank you for the response. I will keep my score.
> >
> > Regarding the local generalization assumption: I appreciate the additional discussion of the intuition behind this, but what I was really looking for in my original comment was a *formal* lower bound showing that there is a sense in which this definition is fundamental.
> >
> > I agree that the generalization from Euclidean space to a metric is not particularly interesting. I think what is more interesting is to understand whether one can hope for a version of the local generalization assumption that does not require any structure of this type.

---

> > > ### Author Response · Authors · 2024-11-24
> > > **New formal lower bound for local generalization**
> > >
> > > Thank you for the response, and we apologize for misunderstanding your original question about local generalization. **We are happy to say that we have proved a new formal lower bound showing the necessity of local generalization**. This was fairly straightforward, since without local generalization, we can set $\mu(s,a) = \mathbf{1}(a = \pi^m(s))$. This means that any single mistake the agent makes is guaranteed to cause catastrophe, so we only need to show that the agent is likely to be make at least one mistake.
> > >
> > > The following theorem shows that even when the input is $\sigma$-smooth and the policy class has finite VC dimension, avoiding catastrophe is impossible without local generalization. In other words, local generalization is necessary for our positive result to hold.
> > >
> > > _Theorem.
> > > Let $S = [0,1]$ and $A = $ {$0,1$}. Assume each state is sampled i.i.d. from the uniform distribution on $S$ and define the mentor policy class by the set of intervals within $S$, i.e., $\Pi = ${$\pi: \exists x,y \in [0,1] \text{ s.t } \pi(s) = \mathbf{1}(s \in [x,y])$}. Then without the local generalization assumption, any algorithm with sublinear queries satisfies_ $\lim_{T\to\infty} \sup_{\mu, \pi^m} \mathbb{E}[R_T] = 1$.
> > >
> > > We are preparing a revised PDF which includes this new result along with the formal proof. We agree that this result strengthens the paper, and we thank the reviewer for suggesting it.
> > >
> > > Of course, this result does not rule out the possibility that local generalization could be replaced by some other assumption. However, we have now both justified local generalization on a conceptual level and showed its necessity on a technical level, which we believe adequately justifies it for the purpose of the paper. We do think it would be interesting to investigate alternatives to local generalization in future work, though.
> > >
> > > Let us know if this addresses your concerns, or if you have further questions.

---

> > > > ### Author Response · Authors · 2024-11-27
> > > > **Second paper revision available, contains proof for new local generalization lower bound**
> > > >
> > > > We would like to alert the reviewer that we have now uploaded the second paper revision, which contains the new local generalization lower bound and its formal proof (Theorem E.3). The result is discussed on lines 526-529 and the formal statement and proof can be found on lines 1624-1736. Please let us know if this doesn't resolve your concerns or if you have any further questions.

---

### Official Review · Reviewer_FhGj · 2024-11-02

**Soundness:** 3
**Presentation:** 3
**Contribution:** 2
**Rating:** 6
**Confidence:** 3

**Summary:**

This paper proposes a new model of online learning, with the performance measured in terms of the product of the per-step value (probabilities). Under this new setting, the authors provide both difficulty results and an algorithm for VC/Littlestone classes.

**Strengths:**

I believe it makes much sense to formulate "asking for help" in terms of query complexity. It is clearly of importance to understand how access to expert advice helps online learners. The proposed algorithm, though looks standard, may also be of techical interest,

**Weaknesses:**

I am a little bit skeptical of the formulation of regret.

1. The probabilistic interpretation of the product $\prod_{t=1}^T \mu(s_t,a_t)$ is unclear. Given the observed sequence of $(s_1,a_1,...,s_T,a_T)$, the product is the probability of avoiding catastrophe only if the catastrophe at step t does not affect the s_{t+1}.

2. For the regret to be non-trivial, it is necessary that $\sum_{t=1}^T -\log \mu^m(s_t)$ to be bounded by a constant as T tends to infinity. I believe this is a very strong assumption on the mentor, and the hence the upper bound can be trivial for most scenarios. It is possible to consider a less restrictive setting? The hardness result may not be an excuse here.

3. It looks possible to take a logarithm and convert this problem to the standard no-regret learning setting, with the goal of achieving sub-constant regret. Given that the mentor needs to achieve constant loss, this goal seems to be achievable. It would be beneficial to include a discussion on why it is infeasible.

**Questions:**

In Sec 3, Q_T is defined to be a deterministic quantity of the algorithm. But all the bounds are in terms of E[Q_T].

---

> ### Author Response · Authors · 2024-11-14
> **Response to Reviewer FhGj**
>
> We thank the reviewer for helpful comments. We respond to each of the weaknesses and also to the reviewer’s question.
>
> **Weaknesses**:
> 1. The formal definition of $\mu(s_t, a_t)$ is actually the chance of catastrophe at time $t$ conditioned on no prior catastrophe (line 263). (Note that the probability of catastrophe at time $t$ conditioned on prior catastrophe already occurring is irrelevant, since the agent has already failed.) Formally, if $E_t$ is the event that catastrophe is avoided at time $t$, then the chain rule of probability implies that $$\Pr[\text{no catastrophe ever}] = \Pr[\cap_{t=1}^T E_t] = \prod_{t=1}^T \Pr[E_t \mid \cap_{k=1}^{t-1} E_k] = \prod_{t=1}^T \mu(s_t, a_t)$$ We will update the paper to better explain the role of this conditioning.
>
> 2. We have two responses here:
>
>     (A) If $\sum_{t=1}^T - \log \mu^m(s_t)$ is not bounded by a constant, this implies that $\prod_{t=1}^T \mu^m(s_t)$ goes to 0. In other words, the environment is sufficiently dangerous and/or the mentor is sufficiently fallible that the mentor’s own policy is guaranteed to eventually cause catastrophe. In such cases, perhaps the task should be avoided altogether; we are imagining applications where the mentor does know a reasonably safe policy.
>
>     (B) The reviewer is correct that subconstant regret becomes trivial if $\sum_{t=1}^T - \log \mu^m(s_t)$ is unbounded and thus $\prod_{t=1}^T \mu^m(s_t)$ goes to 0, but our results say more about the regret than simply that it is subconstant: we also provide rates of convergence. In other words, even if the mentor is guaranteed to eventually cause catastrophe, we can estimate the time horizons on which we can expect the agent to still be relatively safe. If the reviewer recommends this, we could emphasize our convergence rate bounds more in the introduction.
>
>
> 3. The reviewer is roughly correct that taking a logarithm can convert our multiplicative objective into an additive objective. Without help from a mentor, subconstant regret (additive or multiplicative) is impossible because the agent essentially must guess on the first time step, so the agent is wrong with constant probability (and that’s just the first time step). With sublinear queries to a mentor, subconstant additive regret is indeed achievable: we actually prove this as an intermediate step to our main result (lines 1050-1054). To our knowledge, this is a novel result in online learning, which we could also highlight if the reviewer recommends.
>
>     Also, even after the transformation to an additive objective, our problem remains harder than standard online learning. This is because we only receive feedback from queries (whose rate tends to 0 as $T \to \infty$), while online learning typically assumes that payoffs are observed on every time step.
>
> **Question**:
>
> $Q_T$ is in fact a random variable since $a_1,\dots,a_T$ are random variables. We will add an explicit mention of this to avoid confusion.

---

> > ### Author Response · Authors · 2024-11-18
> > **Revised PDF now available**
> >
> > We have now uploaded a revised version of the paper which incorporates the feedback from all reviewers. We believe that this revision addresses each of the questions/weaknesses raised by Reviewer FhGj. Changes are highlighted in red. The relevant changes for this reviewer's feedback can be found on the following line numbers in the revised PDF:
> >
> > W1: Clarified the relationship between the product of payoffs and probability of catastrophe (lines 064-066, see also the unchanged footnote 8)
> >
> > W2: Added a discussion of the quality of the mentor (lines 262-273)
> >
> > W3: Mentioned that we prove a subconstant additive regret bound as an intermediate step (lines 114-115)
> >
> > Q1: Mentioned explicitly that $Q_T$ is a random variable on line 268
> >
> > If the reviewer feels that not all of their concerns have been adequately addressed, we would welcome further discussion.

---

> > ### Comment · Reviewer_FhGj · 2024-11-19
> >
> > Thank you for the responses. Some comments:
> >
> > > $Q_T$ is in fact a random variable
> >
> > Indeed the results seem to be stated for the "intuitive" $Q_T$ defined at line 230. However, in the subsequent paragraph, an alternative definition of the quantity Q_T is proposed, which is deterministic (as it takes supremum over "all possible realizations of the agent’s randomization"). I think this can be confusing.
> >
> > > Regarding additive regret after taking logarithmic
> >
> > Stating the relation between the results of this paper and the logarithmic additive regret can be helpful. I believe a sub-constant additive regret with queries is also an interesting result that is worth spelling out.
> >
> > I will raise my score to 6.

---

> > > ### Author Response · Authors · 2024-11-20
> > >
> > > Thank you for the follow-up and additional feedback. We better understand the confusion with $Q_T$ now. We will remove the second definition of $Q_T$ and clarify the meaning of "$Q_T \in o(T)$". We will also expand the discussion of our subconstant additive regret result and the connections to additive logarithmic regret, as suggested.

---

### Official Review · Reviewer_Qo6Q · 2024-11-02

**Soundness:** 3
**Presentation:** 2
**Contribution:** 2
**Rating:** 6
**Confidence:** 3

**Summary:**

The submission studies an online disaster avoidance problem. The learning agent is allowed to query the mentor to acquire information to avoid disaster. To circumvent the tractability issue of Bayesian methods, a hedge-based algorithm is proposed under an additional assumption called “local generalization.” Also, unlike conventional online learning, the regret is defined in a multiplicative way, and the proven regret bound (Theorem 5.2) is subconstant. Besides the positive result, the submission also shows an impossibility result (Theorem 4.1) in a general case.

**Strengths:**

- A non-Bayesian approach to address the online disaster avoidance problem.
- The analysis simultaneously controls the regret and the query frequency.
- A novel packing argument is developed to bound the query complexity.

**Weaknesses:**

- (1) It seems to be too advantageous for the learning agent to be able to query the optimal policy (i.e., the mentor), given that a local generalization assumption has been made for exploitation. Practically speaking, if querying is allowed, the learning agent should also have to find out who the optimal target to query in the policy class is.
- (2) The argument (lines 508–514) for the novel packing (line 460) is not clear enough. An explanation of the difficulty and the technical contribution is needed.
- (3) There are confusing sentences hindering the readability. These are the sentences in line 126 (As a corollary, …) and line 134 (We initially …).

**Questions:**

- (1) Are there other impossibility results besides Theorem 4.1, especially in the Bayesian regime? If the submission provides the first impossibility result, that is a contribution, too. If there are other impossibility results, we should list them and compare them with Theorem 4.1.
- (2) Is the multiplicative objective really a good choice? For example, consider the following two payoff sequences: (0, 0.999, 0.999, 0.999) and (0.1, 0.1, 0.1, 0.1). The total payoff (multiplicative) of the first sequence is 0 and is smaller than that of the second one. But intuitively, the second sequence may have a better chance of avoiding a disaster.

---

> ### Author Response · Authors · 2024-11-14
> **Response to Reviewer Qo6Q**
>
> We thank the reviewer for helpful comments. We respond to each of the questions and weaknesses stated by the reviewer.
>
> **Questions**:
> 1. We are not aware of other impossibility results, including in the Bayesian regime (except for the trivial result that the problem is hopeless without some sort of help, since the agent has no way to prevent the first action it takes from causing catastrophe). We do believe that our impossibility result is a significant contribution, and we will make this clearer when revising. Thank you for pointing this out.
> 2. Recall that the payoff is the chance of *avoiding* catastrophe that round (conditioned on no prior catastrophe), not the chance of *causing* catastrophe. Under this definition, the chain rule of probability implies that the product of payoffs is exactly the overall chance of avoiding catastrophe. Our goal is to _maximize_ that product. If the agent has 0% chance of avoiding catastrophe on the first round, then catastrophe is in fact guaranteed. This is worse than the second sequence, which retains a nonzero probability of avoiding catastrophe. Correspondingly, $0 \cdot 0.999^3 < 0.1^4$.
>
> **Weaknesses**:
> 1. We first remind the reviewer that we do not assume the mentor to be optimal. More importantly, we are motivated by applications in which there exists an actual human supervisor (or potentially a more powerful AI model) who can respond to queries. Examples include a human doctor supervising AI doctors, a robotic vehicle with a human driver who can take over in emergencies, autonomous weapons with a human operator, and many more. In fact, we suggest that it is crucial for any high-stakes AI applications to have a designated supervisor who can be asked for help. In such applications, the target for queries is known to the agent.
>
>     Also, if we understand correctly what the reviewer means by “find out who the optimal target to query in the policy class is”, this essentially makes queries useless. For example, suppose the policy class contains only the mentor policy and its complement: the agent has no way to know which is which and thus has no way to know which policy it should query. Let us know if we have misunderstood.
>
> 2. We will flesh out this discussion in the paper and are also happy to briefly elaborate here. The technical challenge is that vanilla packing arguments do not consider the actions of the algorithm. Our overall proof requires a set of queries which cover the observed state space with respect to the realized actions of the algorithm, and our technical contribution is to provide a novel method to deal with this scenario. Our technique may be useful for other contexts where a more refined packing argument is needed than one which deals with all data points in aggregate.
>
> 3. Thank you for this feedback. We will address these issues when revising.

---

> > ### Comment · Reviewer_Qo6Q · 2024-11-17
> >
> > Thank you for replying. I still have the following question.
> >
> > W1:
> > The mentor (i.e., the query target) should satisfy a certain "goodness" in the problem. Otherwise, it is easy to prove a good regret bound for a stupid mentor. For instance, no one will choose a target bound for the catastrophe as a mentor. Thus, how does the submission define the "goodness" of a supervisor w.r.t. the problem?

---

> > > ### Author Response · Authors · 2024-11-17
> > >
> > > Thank you for the follow-up. One can define the "goodness" of the mentor as the mentor's expected overall probability of avoiding catastrophe, i.e., $\mathbb{E}\big[\prod_{t=1}^T \mu^m(s_t)\big]$. At minimum, this quantity should be nonzero, or else the regret bound becomes trivial. However, the value of the regret bound certainly increases with the "goodness" of the mentor. In particular, we think that high-stakes AI applications should ensure the presence of a mentor who is almost always safe, i.e., $\mathbb{E}\big[\prod_{t=1}^T \mu^m(s_t)\big] \approx 1$. We will add a discussion of this to the paper.
> > >
> > > It is also worth noting that our formulation of regret with respect to the mentor aligns with prior work on asking for help in the context of Bayesian inference ([8; 9; 25]).
> > >
> > > Does this answer your question? We are happy to discuss further if not.

---

> ### Author Response · Authors · 2024-11-18
> **Revised PDF now available**
>
> We have now uploaded a revised version of the paper which incorporates the feedback from all reviewers. We believe that this revision addresses each of the questions/weaknesses raised by Reviewer Qo6Q. Changes are highlighted in red. The relevant changes for this reviewer's feedback can be found on the following line numbers in the revised PDF:
>
> Q1: Clarified the lack of other impossibility results (lines 171-172)
>
> Q2: Clarified the relationship between the product of payoffs and probability of catastrophe (lines 064-066)
>
> W1: Added justification for the existence of a mentor known to the agent (lines 055-058)
>
> W2: Expanded the description of the novel packing argument (lines 511-515)
>
> W3: Improved the phrasing of the sentences on lines 126 and 134.
>
> See also lines 262-273 for a discussion of the "goodness" of the mentor.
>
> Please let us know if this adequately addresses your concerns. If not, we would welcome further discussion.

---

> > ### Comment · Reviewer_Qo6Q · 2024-11-26
> >
> > Thank you. I have no further questions. I will keep my score for now.

---

### Author Response · Authors · 2024-11-27
**Second paper revision and summary of changes**

We thank all reviewers for a constructive discussion. To recap, our paper proposes a new online learning problem which models the goal of avoiding catastrophe. Conceptually, we show that if a policy class is learnable in the standard online setting (without catastrophic risk), it remains learnable even with catastrophic risk if the agent can occasionally ask for help and the payoff function satisfies local generalization.

The feedback from reviewers has strengthened both the technical content and the framing of our paper. We have now uploaded a second revision, which contains all of the changes from the first revision, plus several additional changes. Content changes are highlighted in red, while terminology/notation/phrasing changes are not highlighted in order to avoid clutter. We have also listed all changes below (first the new changes, followed by a full list of all changes). Note that the line numbers in our previous comments about the first revision (titled “revised PDF now available”) may no longer line up with this new revision.

**New changes added after first revision:**
1. _[Reviewer kZKp W1]_ As requested, we proved a new lower bound showing that the local generalization assumption is fundamental and included it as Theorem E.3. This result is discussed on lines 526-529 and the formal statement and proof can be found on lines 1624-1736. **The new local generalization lower bound is the most significant change in the second revision.**
2. _[Reviewer FhGj W3]_ Made explicit the subconstant regret bound we prove for the standard additive regret (lines 443-450 for the statement of Theorem 5.3, lines 135-140 for the discussion).
3. _[Reviewer FhGj Q1]_ Clarified the definition of $Q_T$ (lines 229-234). We also added a new result (Proposition E.2 on lines 1590-1622, mentioned by footnote 7) which shows that the tractability of our problem does not depend on whether we require sublinear expected queries or sublinear worst-case queries.
4. _[Reviewer kZKp Concern 6]_ To better align with existing online learning literature, we switched the term “state” to “input” and switched the notation $s_t$ and $a_t$ for states and actions to $x_t$ and $y_t$ for inputs and actions.

Below we provide an exhaustive list of all changes, organized by reviewer:

**Reviewer Qo6Q:**

_[W1]_ Added justification for the existence of a mentor known to the agent (lines 055-058).

_[W2]_ Expanded the description of the novel packing argument (lines 509-513).

_[W3]_ Improved the phrasing of sentences on lines 126 and 134.

_[Q1]_ Clarified the lack of other impossibility results (lines 171-172).

_[Q2]_ Clarified the relationship between the product of payoffs and probability of catastrophe (lines 065-067, also preexisting footnote 8).

_[Follow up question]_ Added discussion of the "goodness" of the mentor (lines 261-272).

**Reviewer FhGj:**

_[W1]_ Clarified the probabilistic interpretation of the product of payoffs (lines 065-067, also preexisting footnote 8).

_[W2]_ Added discussion of the quality of the mentor (lines 261-272).

_[W3]_ Made explicit the subconstant regret bound for standard additive regret (discussed under “New changes…”) and added comparison to online learning with log loss (lines 192-196).

_[Q1]_ Discussed under “New changes…”.

**Reviewer kZKp:**

_[Concern 1]_ Expanded justification for the value of work in the context of AI risk (lines 055-059).

_[Concern 2]_ Improved the discussion justifying local generalization (lines 238-248) and added a new formal lower bound (discussed under “New changes…”).

_[Concern 3]_ Mentioned that our assumption of Euclidean space is only for convenience (footnote 5).

_[Concern 4]_ Added discussion of online learning with log loss (lines 192-196).

_[Concern 5]_ Added discussion of imitation learning under related work (lines 206-207).

_[Concern 6]_ Discussed under “New changes…”.

**Reviewer E2Pc:**

_[W1]_ Expanded the description of the novel packing argument (lines 509-513).

_[W2]_ Added comparison to CMDP literature, particularly Liu et al. [2021] and Stradi et al. [2024] (lines 173-181).

We believe that our changes address all of the concerns raised by reviewers and significantly strengthen the paper in general.

---

### Meta-Review · Area_Chair_5LTh · 2024-12-23

**Metareview:**

This is a borderline paper with potential. However, there is also criticism and there is a feeling that the paper is not quite there yet. This concerns, in particular, links to the AI risk literature and, more importantly, the local generalization assumption. I believe that the paper needs more work before being ready for publication.

**Additional Comments On Reviewer Discussion:**

There was quite a bit of a useful discussion between the authors and reviewers. Though the scores did not change after the discussion.

---

### Decision · Program_Chairs · 2025-01-22

Reject